# WSX1 act as a tumor suppressor in hepatocellular carcinoma by downregulating neoplastic PD-L1 expression

Man Wu[1,2], Xueqing Xia[2], Jiemiao Hu [2], Natalie Wall Fowlkes [3] & Shulin Li [2✉]

WSX1, a receptor subunit for IL-27, is widely expressed in immune cells and closely involved in immune response, but its function in nonimmune cells remains unknown. Here we report that WSX1 is highly expressed in human hepatocytes but downregulated in hepatocellular carcinoma (HCC) cells. Using *NRAS/AKT*-derived spontaneous HCC mouse models, we reveal an IL-27–independent tumor-suppressive effect of WSX1 that largely relies on CD8$^+$ T-cell immune surveillance via reducing neoplastic PD-L1 expression and the associated CD8$^+$ T-cell exhaustion. Mechanistically, WSX1 transcriptionally downregulates an isoform of PI3K—PI3Kδ and thereby inactivates AKT, reducing AKT-induced GSK3β inhibition. Activated GSK3β then boosts PD-L1 degradation, resulting in PD-L1 reduction. Overall, we demonstrate that WSX1 is a tumor suppressor that reinforces hepatic immune surveillance by blocking the PI3Kδ/AKT/GSK3β/PD-L1 pathway. Our results may yield insights into the host homeostatic control of immune response and benefit the development of cancer immunotherapies.

[1] Liver Cancer Institute & Key Laboratory of Carcinogenesis and Cancer Invasion, Zhongshan Hospital, Fudan University, Shanghai 200032, PR China. [2] Department of Pediatrics-Research, The University of Texas MD Anderson Cancer Center, Houston, TX 77030, USA. [3] Department of Veterinary Medicine and Surgery, The University of Texas MD Anderson Cancer Center, Houston, TX 77030, USA. ✉email: sli4@mdanderson.org

Hepatocellular carcinoma (HCC) is one of the most common malignancies worldwide and ranks as the fifth leading cause of cancer-related mortality[1–3]. The development of HCC is a multifactorial process in which genetic and epigenetic alterations in regulatory genes result in the activation of oncogenes and inactivation of tumor suppressor genes[4]. Unlike most other malignancies, HCC is an inflammation- and immune-related tumor that arises almost exclusively in an inflamed fibrotic or cirrhotic setting. Significantly, a tightly controlled immunological network is designed to detect and eliminate transformed cells, but this process is frequently dysregulated in HCC[5]. Recent breakthroughs in immunotherapy targeting immune checkpoints have substantially improved HCC patient survival[6,7], further supporting the crucial contribution of immune dysregulation to HCC development. However, so far, the mechanism by which HCC cells induce immunotolerance and escape from host homeostatic immunosurveillance remains unclear.

Programmed death ligand-1 (PD-L1) is a critical immune checkpoint protein whose engagement with its receptor, programmed death 1 (PD-1), on T cells activates co-inhibitory signaling to suppress effector T-cell function[8,9]. The physiological role of the PD-L1/PD-1 axis is maintaining the balance between peripheral tolerance and autoimmunity, but cancer cells hijack this process to escape from host immune surveillance[10]. Cancer cells dodge immune elimination through their expression of PD-L1, which interacts with PD-1 expressed on T cells to induce immunodepression[11,12]. Over the past decade, blockade of the PD-L1/PD-1 axis has shown remarkable clinical responses in a variety of advanced cancers, including HCC[6,7,13]. Nivolumab and durvalumab, inhibitors targeting PD-L1/PD-1, yielded promising outcomes in phase II clinical trials and were granted accelerated approval by the US Food and Drug Administration as a second-line treatment for HCC[6,13–15]. These encouraging clinical benefits highlight the critical role of the PD-L1/PD-1 axis in HCC pathogenesis. In physiological conditions, PD-L1 expression is under stringent control to prevent immune overreaction while maintaining a proper level of immune defense. However, this control system is often destroyed in cancer, resulting in PD-L1 dysregulation, which subsequently initiates PD-L1/PD-1 axis–mediated immune evasion and promotes tumor formation. Nonetheless, little is known about the mechanism underlying the homeostatic control of PD-L1 expression and its dysregulation in cancer. Understanding the multifaceted control of PD-L1 would facilitate the development of more effective therapeutic strategies.

WSX1, homologous to the IL-12 receptor β2 chain, is a class I cytokine receptor, which consists of a single transmembrane domain, a WSX1 signature motif, a box 1 motif in its intracellular region, and 7 potential N-glycosylation sites in its extracellular domain[16]. WSX1, together with gp130, constitutes a functional signal-transducing receptor for IL-27; lack of either subunit attenuates IL-27-mediated signaling[17]. WSX1 was previously thought to be expressed exclusively in immune cells, mediating IL-27-dependent pro-inflammatory or anti-inflammatory immune responses[17,18]. However, we and others previously demonstrated that WSX1 is also expressed in multiple types of epithelial tumor cells, including breast tumors, melanomas, and lung carcinomas[19–21]. The presence of WSX1 sensitized IL-27-independent natural killer (NK) cell-mediated antitumor surveillance in breast tumors[20] and inhibited melanoma growth in an IL-27-dependent manner[19]. In addition, a recent study reported that WSX1 deficiency in mice promoted the oncogenic properties of mutant p53[22], indicating a tumor-suppressive role of WSX1. These published biological functions of WSX1 largely depend on the presence of its ligand IL-27.

In this study, we reveal an IL-27-independent biological function of WSX1 as a potential HCC suppressor, a function that relies primarily on WSX1's regulation of CD8+ T cell-mediated adaptive immunity. In brief, we use multiple human tissue microarrays, spontaneous HCC mouse models, and current immune and molecular profiling tools to show that WSX1 downregulates neoplastic PD-L1 and reduces CD8+ T-cell exhaustion in the tumor microenvironment, leading to inhibition of oncogene-mediated HCC formation.

## Results

**WSX1 is highly expressed in human normal liver tissues, and its downregulation in HCC closely correlated with poor prognosis.** While the role of WSX1 has been well studied in the immune system, its expression pattern and biological function in nonimmune cells remains unknown. To clarify the physiological distribution of WSX1, we first determined WSX1 expression using a human multiple normal tissue microarray (FDA662a) with 33 different types of human normal organ tissue. In concurrence with previous reports[16–18], positive expression of WSX1 was found in immune cell-enriched tissues such as spleen and lymph node, and reached a much higher level in thymus gland and bone marrow. Unexpectedly, we observed high expression of WSX1 in multiple normal human organs, such as colon, intestine, and kidney (Supplementary Fig. 1). Notably, normal human liver tissues exhibited evenly positive staining for WSX1 (Fig. 1a), most of which was uniformly distributed in the cytoplasm and cell membrane of hepatocytes. In addition, the expression level of WSX1 in liver tissues was even higher than that in immune cell-enriched tissues. Combined with our previous discoveries that WSX1-deficient mice were highly sensitive to liver inflammation[23,24], we speculate that WSX1 plays a critical role in the liver.

To validate the above results and explore whether WSX1 is associated with HCC development, we further analyzed WSX1 expression in human liver tissue microarrays (BC03116a and HLiv-HCC180Sur-03), which included 130 cases of HCC, 17 normal tissue samples from healthy donors, and 103 normal tumor-adjacent liver tissues (NAT). The clinical and pathological characteristics of all 130 HCC patients are shown in Table 1. Among the 130 HCC cases, relative survival follow-up information was available for 90 patients (HLiv-HCC180Sur-03). Since positive-staining areas exhibited similar intensity, results were quantified as the percentage of WSX1+ area by ImageJ software. We consistently observed an evenly diffused positive staining pattern for WSX1 in normal liver tissues (mean ± SD, 48.97% ± 6.64%), mostly in hepatocytes. However, WSX1 expression was significantly lower in NAT (39.31% ± 8.58%, $P < 0.0001$) and was the lowest in HCC tissues (14.31% ± 7.95%, $P < 0.0001$, Fig. 1b). Additionally, using the median value as a cutoff, the 90 HCC patients in HLiv-HCC180Sur-03 were manually divided into low ($n = 47$) and high ($n = 43$) WSX1 expression groups. As expected, patients with lower WSX1 expression had shorter overall survival (hazard ratio [HR] = 2.460, $P = 0.0034$, Fig. 1c). Furthermore, we observed that WSX1 expression was lower in histological grade III (10.31% ± 5.18%, $n = 34$), compared to its expression in grade I (19.51% ± 8.67%, $n = 21$, $P < 0.0001$) and grade II (14.80% ± 7.97%, $n = 73$, $P = 0.0123$). No significant association of WSX1 expression with age, sex, or TNM stage was found (Table 1). Collectively, these results suggested that WSX1 plays a negative role in HCC development.

**WSX1 retards HCC development in vivo independently of IL-27.** To clarify the role of WSX1 in HCC development, we established a spontaneous HCC mouse model in immune-competent FVB/NJ mice. Consistent with previous studies[25,26],

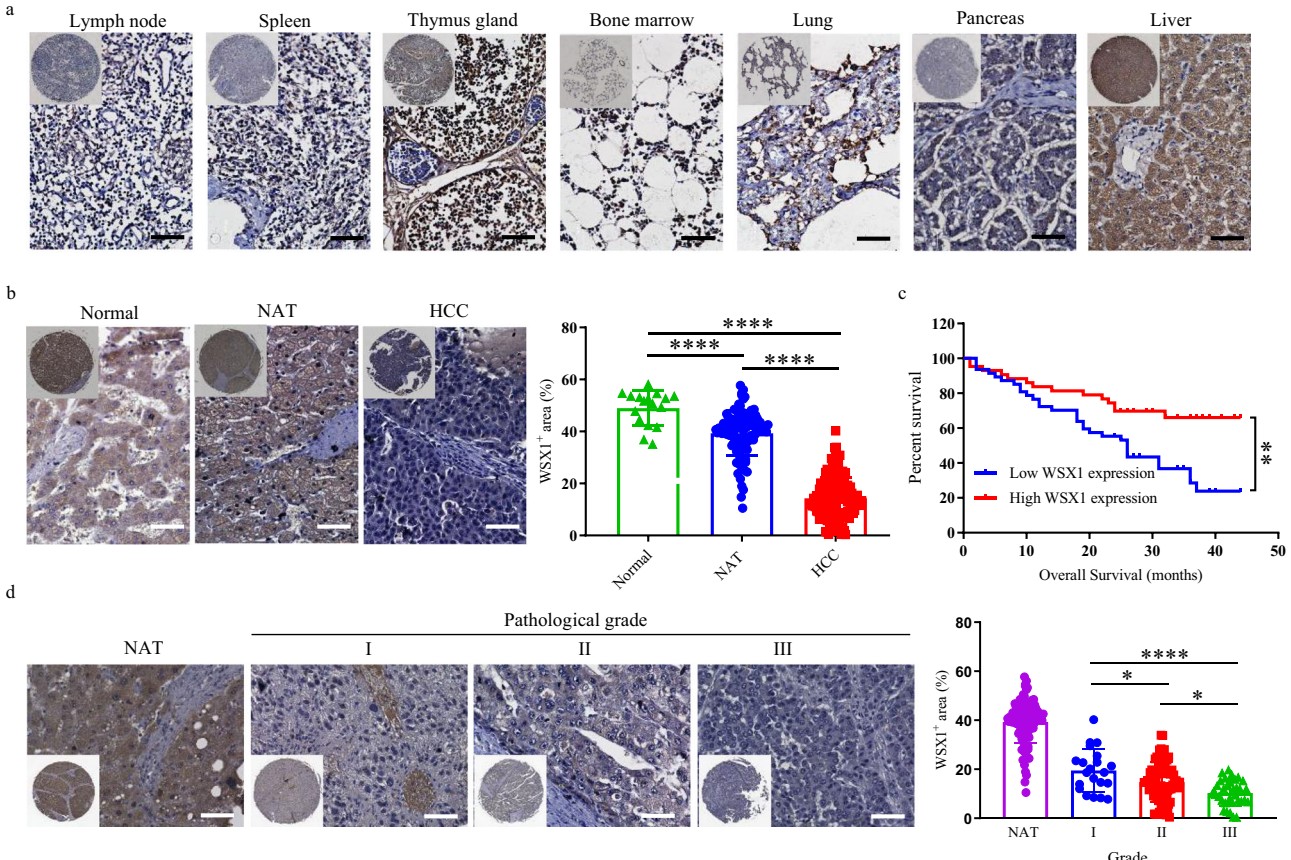

**Fig. 1 WSX1 is highly expressed in normal hepatocytes, and its downregulation in HCC correlated with poor prognosis. a** Immunohistochemical (IHC) staining of WSX1 in a human normal tissue microarray (TMA, FDA662a). The images shown are representatives for results of 2 individuals. **b** IHC staining of WSX1 in HCC TMAs (BC03116a and HLiv-HCC180Sur-03), including human normal liver tissue ($n = 17$), NAT ($n = 103$, $P < 0.0001$ compared to normal liver tissue), and HCC ($n = 130$, $P < 0.0001$ compared to both normal liver tissue and NAT) samples. Statistical analysis results are based on quantification of the percentage of WSX1$^+$ area in each TMA tissue core. **c** Comparisons of the overall survival of HCC patients between low ($n = 47$) and high WSX1 expression group ($n = 43$, $P = 0.0034$). Forty HCC patients in TMA BC03116a were excluded due to lack of survival data. **d** WSX1 expression levels among different tumor pathological grades in HCC patients, including NAT ($n = 103$), grade I ($n = 21$), grade II ($n = 73$, $P = 0.0321$ compared to grade I) and grade III ($n = 34$, $P < 0.0001$ compared to grade I and $P = 0.0123$ compared to gradeII). Quantification and statistical analysis results are shown on the right. Scale bars, 50 μm. Quantitative data are presented as mean ± SD. One-way ANOVA was used to calculate the $P$ values. Tukey-Kramer multiple comparison test was used for pairwise comparisons in the ANOVA analysis. The survival curves were analyzed by the Kaplan–Meier method, and the log-rank test was used to compare overall survival between groups. All statistical tests were two-sided. *$P < 0.05$, **$P < 0.01$, ****$P < 0.0001$. HCC hepatocellular carcinoma, NAT normal tumor-adjacent liver tissue. Source data are provided as a Source Data file.

**Table 1 Clinical and pathological information of 130 HCC patients included in TMAs.**

| Variable | No. of patients | P value |
|---|---|---|
| Age (years) | | 0.5741 |
| <60 | 94 | |
| ≥60 | 36 | |
| Sex | | 0.9238 |
| Male | 105 | |
| Female | 25 | |
| Pathology Grade | | <0.0001* |
| I | 21 | |
| II | 73 | |
| III | 34 | |
| TNM Stage | | 0.8736 |
| 1 | 10 | |
| 2 | 55 | |
| 3 | 56 | |

we found that hydrodynamic injection (HDI) of *NRAS/AKT* oncogenes induced the development of nodular and diffuse HCC within 6 weeks with no exception (Fig. 2a). Hematoxylin and eosin (H&E) staining revealed that neoplastic lesions occupied up to 90% of the liver parenchyma. However, with weekly HDI of plasmid DNA encoding WSX1, HCC formation was significantly retarded, with sporadic distribution of tumor nodules occupying about 20% of the liver parenchyma ($P < 0.0001$, Fig. 2b, c). Consistently, HDI of WSX1 caused remarkable decreases in liver weight ($P = 0.0088$, Fig. 2d), with pronounced survival extension (HR = 0.2046, $P = 0.0217$, Fig. 2e). Moreover, consistent with our results in human HCC tissues, hepatic expression of WSX1 was decreased in oncogene-treated mice compared to untreated mice and was largely recovered following HDI of plasmids encoding WSX1 (Fig. 2k).

WSX1$^{-/-}$ C57BL/6 J mice were also included to establish the spontaneous HCC model, in which half doses of *NRAS/AKT* oncogenes were injected. Consequently, although WSX1$^{-/-}$ mice were generally viable and displayed no overt abnormalities, they

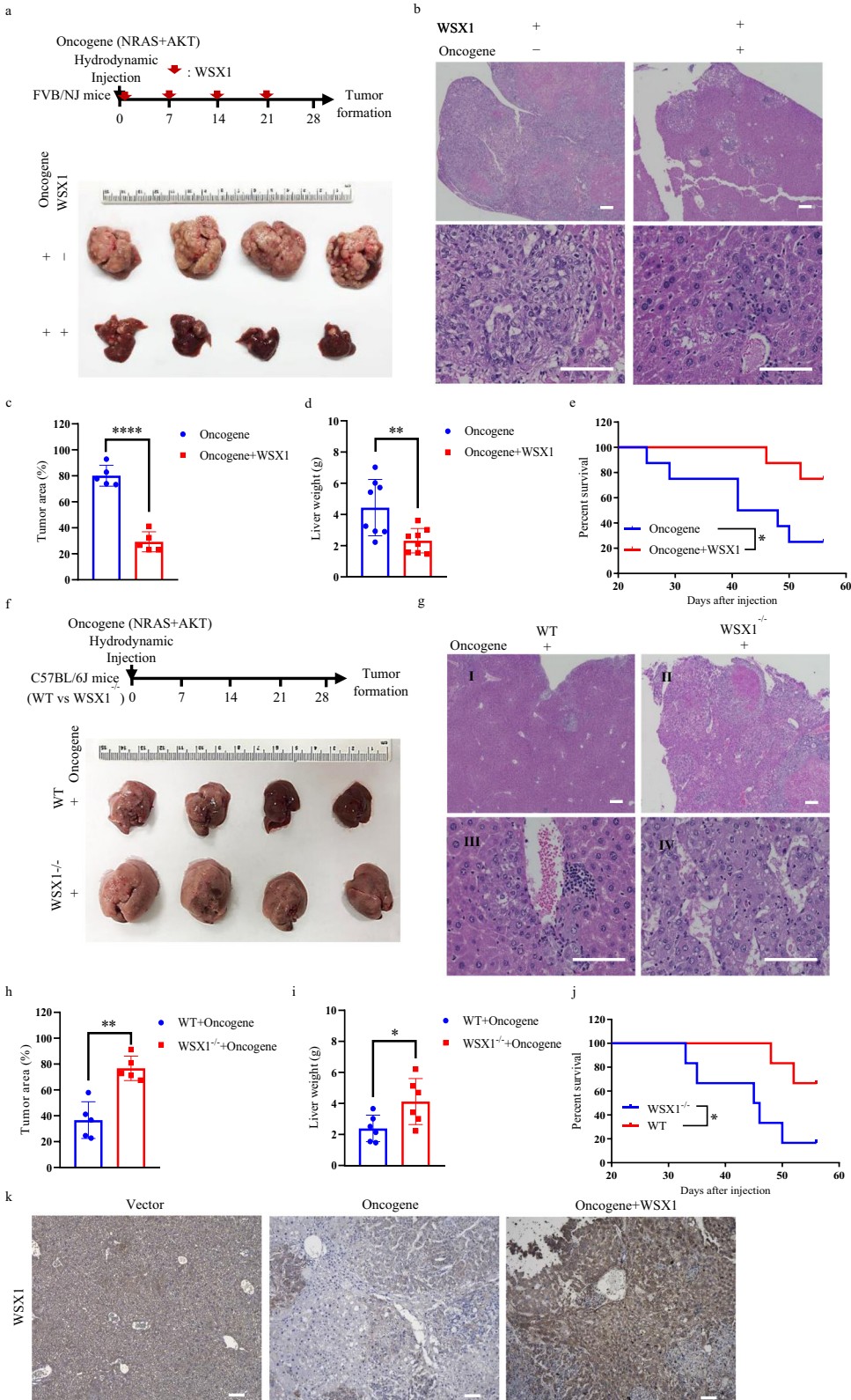

were much more susceptible to *NRAS/AKT*-induced oncogenesis (Fig. 2f). Specifically, compared with wild-type mice, WSX1$^{-/-}$ mice showed earlier tumor formation, more neoplastic lesions ($P = 0.0012$, Fig. 2g, h), increased liver weight ($P = 0.0328$, Fig. 2i), and significantly poorer survival (HR = 4.821, $P = 0.0252$, Fig. 2j). Together, these results further supported the tumor-suppressive role of WSX1 in HCC development.

Furthermore, a previous study reported that the inhibitory effect of WSX1 on melanoma cell proliferation was dependent on the presence of its ligand, IL-27[19]. Therefore, we established a spontaneous HCC mouse model in IL-27p28$^{-/-}$ C57BL/6J mice as well. IL-27p28 is an indispensable component of the heterodimer cytokine IL-27[27]. As a result, we found that IL-27p28 deficiency had no significant influence on the tumor-suppressive effect of WSX1.

**Fig. 2 WSX1 retards oncogenic *NRAS/AKT*-induced HCC development in vivo. a** (Top) Summary of *NRAS/AKT* oncogene-derived spontaneous HCC mouse model in FVB/NJ mice ($n = 8$). Arrowheads represent hydrodynamic injection of WSX1 every week. (Bottom) Representative images of entire mouse livers in the oncogene and oncogene + WSX1 groups. **b** Comparisons of hematoxylin and eosin (H&E) histology. **c** Comparisons of percentage of liver area containing preneoplastic/tumor lesions based on H&E results ($P < 0.0001$). **d** Difference in liver weight between oncogene and oncogene + WSX1 groups ($P = 0.0088$). **e** Comparison of overall survival ($P = 0.0217$). **f** (Top) Summary of HCC mouse model in wild-type or WSX1$^{-/-}$ C57BL/6J mice ($n = 6$). (Bottom) Representative images of entire mouse livers in wild-type or WSX1$^{-/-}$ mice. **g** Comparisons of H&E histology. **h** Comparisons of percentage of liver area containing preneoplastic/tumor lesions ($P = 0.0012$). **i** Difference in liver weight between wild-type or WSX1$^{-/-}$ mice ($P = 0.0328$). **j** Comparisons of overall survival ($P = 0.0252$). **k** Expression of WSX1 in mouse liver tissues in the vector, oncogene, and oncogene + WSX1 groups. Scale bars, 100 μm. All data and images are representative of 3 independent experiments. Quantitative data are presented as mean ± SD and analyzed by two-sided Student $t$ test. The survival curves were analyzed by the Kaplan–Meier method, and the log-rank test was used to compare overall survival between groups. *$P < 0.05$, **$P < 0.01$, ****$P < 0.0001$. Source data are provided as a Source Data file.

In the absence of IL-27, WSX1 still impaired *NRAS/AKT* oncogene-induced HCC formation and improved overall survival (Supplementary Fig. 2), indicating the inhibitory function of WSX1 in HCC is independent of IL-27.

**WSX1 impedes HCC development by maximizing CD8$^+$ T cell-mediated antitumor immunosurveillance.** The crosstalk between tumor cells and the immune system is generally accepted as a pivotal factor in HCC development[5]. Considering that WSX1 had no obvious effect on HCC cell proliferation and migration in vitro (Supplementary Fig. 3b, c), and that WSX1 has long been reported to be closely connected with immunoregulation[16–18], we hypothesized that the immune system might be involved in the tumor-suppressive function of WSX1. To test this hypothesis, we constructed a spontaneous HCC mouse model using immuno-deficient NOD *scid* gamma (NSG) mice (which lack mature T cells, B cells, and NK cells). As expected, WSX1 failed to block *NRAS/AKT* oncogene-induced HCC development in these NSG mice ($P = 0.99$, Supplementary Fig. 4), strongly supporting our notion that an intact immune system is indispensable for the antitumor function of WSX1.

To determine how immunity contributes to the tumor-suppressive function of WSX1, we isolated intrahepatic infiltrating immune cells from entire livers obtained from spontaneous HCC mouse models and then performed cell profiling by time-of-flight mass cytometry (CyTOF). PhenoGraph analysis of 34 cell markers' expression profiles identified 6 main immune cell subsets: T cells, B cells, NK cells, macrophages, dendritic cells, and other CD3$^-$ cells (Fig. 3a). As shown in Fig. 3b, WSX1 had no significant effect on the proportions of these 6 cell subsets among the entire population of intrahepatic immune cells. Interestingly, about 80% of intrahepatic immune cells consisted of T-cells (Fig. 3b). Thus, our further analysis focused on T-cell clusters. As depicted, among the entire intrahepatic T-cell population, the proportions of CD4$^+$, CD8$^+$, CD4$^+$CD8$^+$ double positive (DP), and CD4$^-$CD8$^-$ double negative (DN) T-cells were similar between the oncogene and oncogene + WSX1 groups (Fig. 3c), and CD8$^+$ T-cells constituted up to 50% of all intrahepatic T-cells in both groups. Moreover, in the T-cell panel, PhenoGraph analysis identified 13 T-cell subsets: 2 CD4$^+$ (T2 and T10), 7 CD8$^+$ (T3, T5, T7, T8, T9, T11, T12), 3 DP (T1, T4, T6), and 1 DN (T13) (Fig. 3d). The expression profiles of 28 different markers on each T-cell cluster were visualized in a heatmap (Fig. 3e). Figure 3f shows the proportions of each T-cell cluster among the entire intrahepatic T-cell population. Notably, WSX1 significantly increased the proportions of the T6 (DP T, $P = 0.0003$), T9 (CD8$^+$ T, $P = 0.0016$) and T13 (DN T, $P = 0.0453$) subsets, but reduced the proportions of the T1 (PD-1$^+$LAG-3$^+$ DP T, $P = 0.0114$), T5 (LAG-3$^+$CTLA-4$^{Lo}$CD8$^+$ T, $P = 0.0126$), and T11 (PD-1$^{Lo}$LAG-3$^+$CTLA-4$^+$CD8$^+$ T, $P = 0.0216$, Fig. 3f) subsets. These results indicated that WSX1 treatment predominantly affected CD8$^+$ T cells. Moreover, all

CD8$^+$ T-cell clusters, including the T1 DP T-cell cluster, that were decreased by WSX1 had high- and co-expression of PD-1, CTLA-4, and LAG-3, which are inhibitory receptors of T-cell function[28]. T-cell exhaustion is often characterized by high immune inhibitory receptors co-expression (e.g., PD-1, LAG-3, CTLA-4) and progressive loss of T-cell function, including hierarchical loss of cytokine production (e.g., IL-2, IFN-γ)[29,30]. Recent studies revealed that the HMG-box transcription factor TOX is a central regulator of T-cell exhaustion[31–34].

Our independent multicolor flow cytometry experiments further validated that CD8$^+$ T cells in the oncogene + WSX1 group had lower expression not only of T-cell exhaustion markers (PD-1: $P = 0.0074$; CTLA-4: $P = 0.0471$; LAG-3: $P = 0.0295$) but also of the T-cell exhaustion driver TOX ($P = 0.0116$, Supplementary Fig. 5b, d). In addition, WSX1 increased the expression levels of T-cell functional markers (granzyme B: $P = 0.0344$; perforin: $P = 0.0440$; Ki67: $P = 0.0418$; IL-2: $P = 0.0405$; IFN-γ: $P = 0.0268$; Supplementary Fig. 5c, d). Together, the above results suggested that the presence of WSX1 could relieve CD8$^+$ T-cell exhaustion, which might be responsible for WSX1-mediated suppression of HCC formation and progression.

To validate our hypothesis that the tumor-suppressive effect of WSX1 relies on its impact on CD8$^+$ T-cells, we next performed in vivo depletion of CD8$^+$ and CD4$^+$ T-cells as well as NK cells using specific antibodies. The efficiency of in vivo immune cell depletion is shown in Fig. 4a. Depletion of CD8$^+$ T-cells substantially impaired WSX1-mediated inhibition of HCC development induced by oncogene attack, yielding massive nodular and diffuse liver tumors ($P = 0.0002$, Fig. 4b, c). In contrast, depletion of CD4$^+$ T-cells or NK cells had no significant effect or only mildly increased liver weight. Accordingly, in vivo depletion of CD8$^+$ T-cells almost completely abolished the WSX1-induced survival extension (HR = 7.078, $P = 0.0343$, Fig. 4d).

**WSX1 relieves PD-L1/PD-1 axis-induced T-cell exhaustion by downregulating PD-L1 expression in tumor cells.** Since HDI of plasmid DNA encoding WSX1 had no significant influence on WSX1 expression in intrahepatic CD8$^+$ T cells (Supplementary Fig. 5d), we speculated that the regulation of WSX1 on T-cell exhaustion was mostly mediated by its effect on tumor cells. Considering that the engagement of PD-L1 with its receptor PD-1 on T-cells is the most important reason for T-cell exhaustion[35], we next tested whether WSX1 reduces T-cell exhaustion by affecting PD-L1 expression in tumor cells. First, we used the human HCC cell lines SNU449 and SNU475, which have low expression levels of WSX1, to construct 2 stable cell lines with WSX1 overexpression (449$^{WSX1}$ and 475$^{WSX1}$). We also knocked down WSX1 using a CRISPR/Cas9 guiding RNA (crWSX1) in human HCC cell line SNU398, which has a relatively high expression level of WSX1 (Supplementary Fig. 3a). compared with parental cells, both the cell surface expression and total

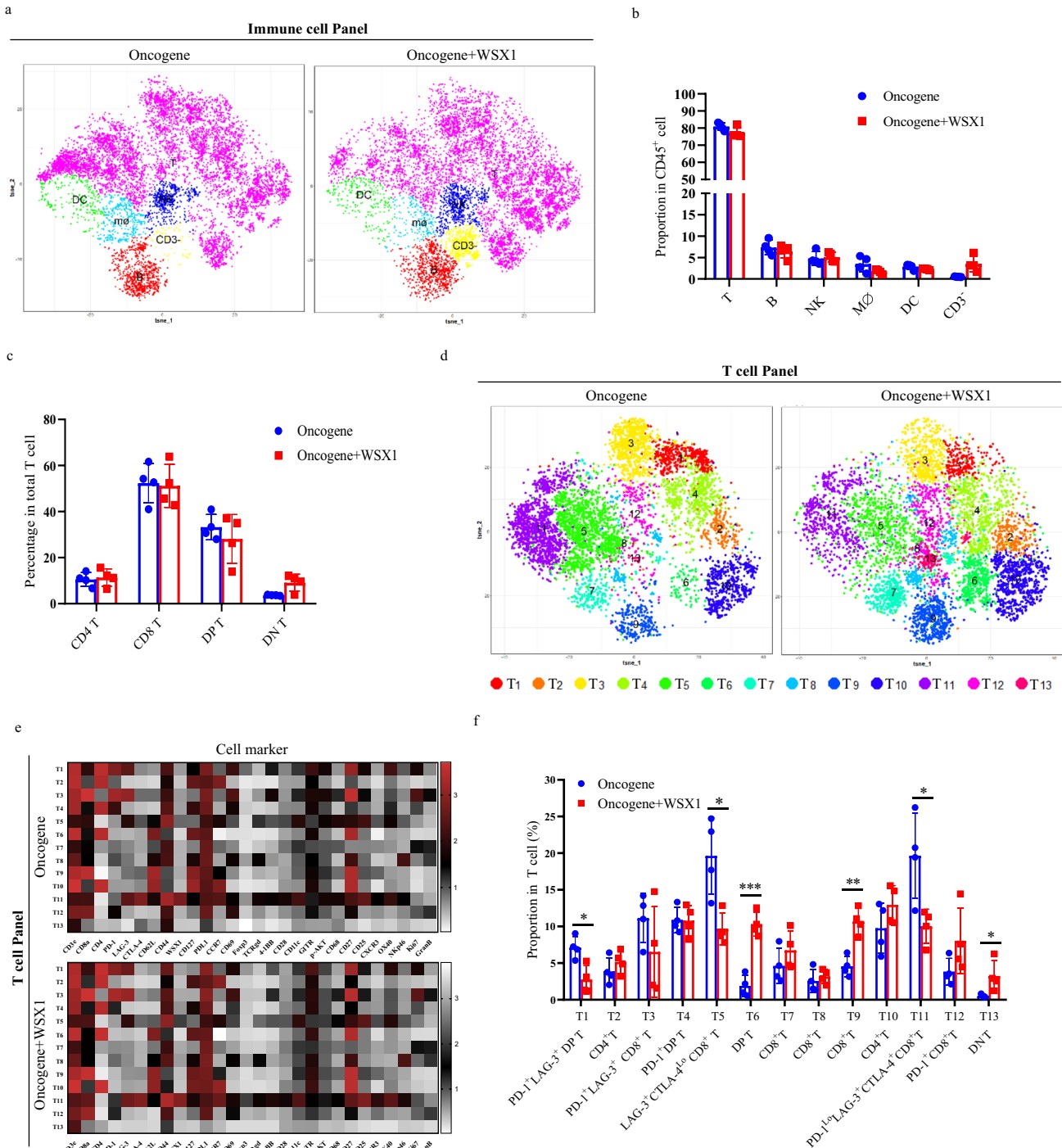

**Fig. 3 WSX1 inhibits HCC development by relieving T-cell exhaustion. a** t-distributed stochastic neighbor embedding (t-SNE) map derived from time-of-flight mass cytometry (CyTOF) analysis of intrahepatic immune cells obtained from the HCC mouse model in Fig. 2a (n = 4). Cells are colored by clusters identified by Rphenograph. Clusters were grouped by expression profile and manually assigned to 6 main cell subsets: T cells, B cells, NK cells, Mϕ, DCs, and other CD3− cells. **b** The percentage of T, B, NK, Mϕ, DC, and other CD3− cells among all intrahepatic immune cells (n = 4). **c** The proportion of CD4+, CD8+, CD4+CD8+ DP, and CD4−CD8− DN T-cell subsets among total intrahepatic T cells (n = 4). **d** t-SNE map of intrahepatic T cells derived from CyTOF analysis. Rphenograph identified 13 T-cell subsets based on expression profiles of 28 markers. **e** Heatmap showing expression of 28 T-cell panel markers in 13 T-cell clusters. **f** Differences in the proportion of 2 CD4+ (T2 and T10), 7 CD8+ (T3, T5, T7, T8, T9, T11, T12), 3 DP (T1, T4, T6), and 1 DN (T13) T-cell clusters among total intrahepatic T cells (n = 4). WSX1 reduced the proportion of T1 (P = 0.0114), T5 (P = 0.0126) and T11 (P = 0.0216), while increased the proportion of T6 (P = 0.0003), T9 (P = 0.0016) and T13 subsets (P = 0.0453). All data are representative of 2 independent experiments. Quantitative data are presented as mean ± SD and analyzed by two-sided Student t test. *P < 0.05, **P < 0.01, ***P < 0.001. NK cells natural killer cells, Mϕ macrophages, DCs dendritic cells, DP double positive, DN double negative. Source data are provided as a Source Data file.

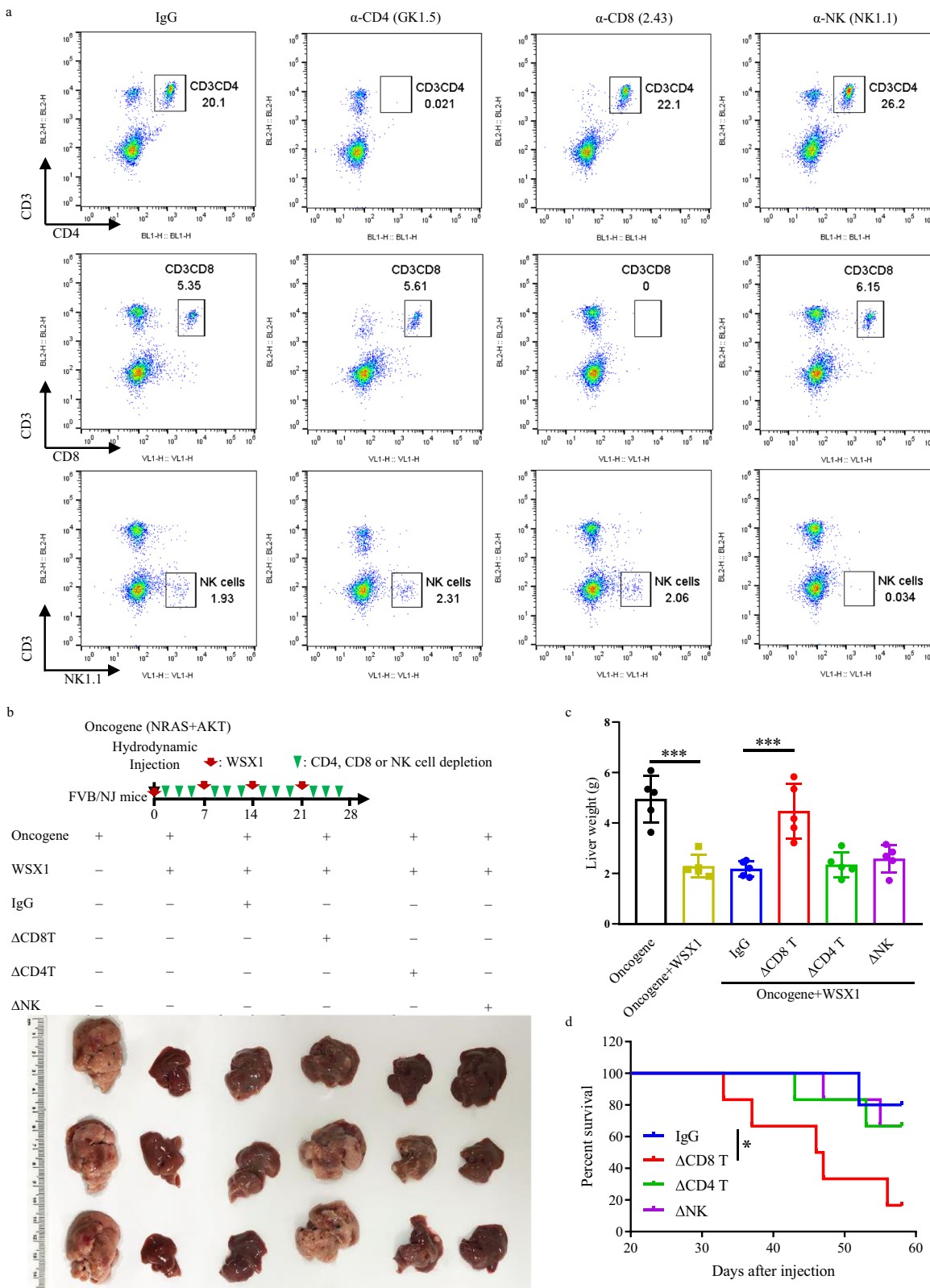

protein level of PD-L1 decreased in WSX1-overexpressing cell lines 449$^{WSX1}$ ($P < 0.0001$) and 475$^{WSX1}$ ($P = 0.0002$, Fig. 5a, b), but increased in WSX1-knockdown cells ($P = 0.0028$, Fig. 5c, d).

To confirm the connection between WSX1 and PD-L1 in vivo, we performed CyTOF analysis of mouse hepatocytes obtained from entire livers in spontaneous HCC mouse models. Due to the distinctive characteristics of the spontaneous HCC model, there are

no visible explicit boundaries at which to distinguish tumor lesions, preneoplastic areas, and normal liver tissues. And no marker is specific enough to distinguish spontaneous tumor cells from normal hepatocytes. Thus, our further studies analyzed hepatocytes as a whole. Consistent with our results in vitro, CyTOF analysis results demonstrated that WSX1 indeed decreased PD-L1 expression in hepatocytes ($P = 0.0012$, Fig. 5e, f), which was further confirmed by

**Fig. 4 WSX1-induced HCC regression is dependent on CD8$^+$ T-cells.** Depletion antibodies against CD8$^+$ T ($\alpha$-CD8), CD4$^+$ T ($\alpha$-CD4), and NK ($\alpha$-NK) cells were used for immune-cell depletion in the spontaneous HCC mouse model. **a** Efficiency of in vivo immune-cell depletion was validated by flow cytometry. The gating strategy for sorting CD8$^+$ T-cells, CD4$^+$ T-cells, and NK cells is shown in the supplementary Fig. 8a. **b** Effect of WSX1 on tumor growth with or without in vivo immune-cell depletion ($n = 5$). Red arrowheads represent WSX1 injection once a week. Green arrowheads represent injection of the indicated antibodies 3 times a week. **c** Depletion of CD8$^+$ T-cells impaired WSX1-mediated inhibition of HCC formation ($P = 0.0002$). **d** Depletion of CD8$^+$ T-cells decreased the WSX1-induced survival extension (HR = 7.078, $P = 0.0338$). All data and images are representative of 2 independent experiments. Quantitative data are presented as mean ± SD and analyzed by One-way ANOVA analysis. Tukey-Kramer multiple comparison tests were used for pairwise comparisons in the ANOVA analysis. The survival curves were analyzed by the Kaplan–Meier method, and the log-rank test was used to compare overall survival between groups. All statistical tests were two-sided. *$P < 0.05$, **$P < 0.01$, ***$P < 0.001$. Source data are provided as a Source Data file.

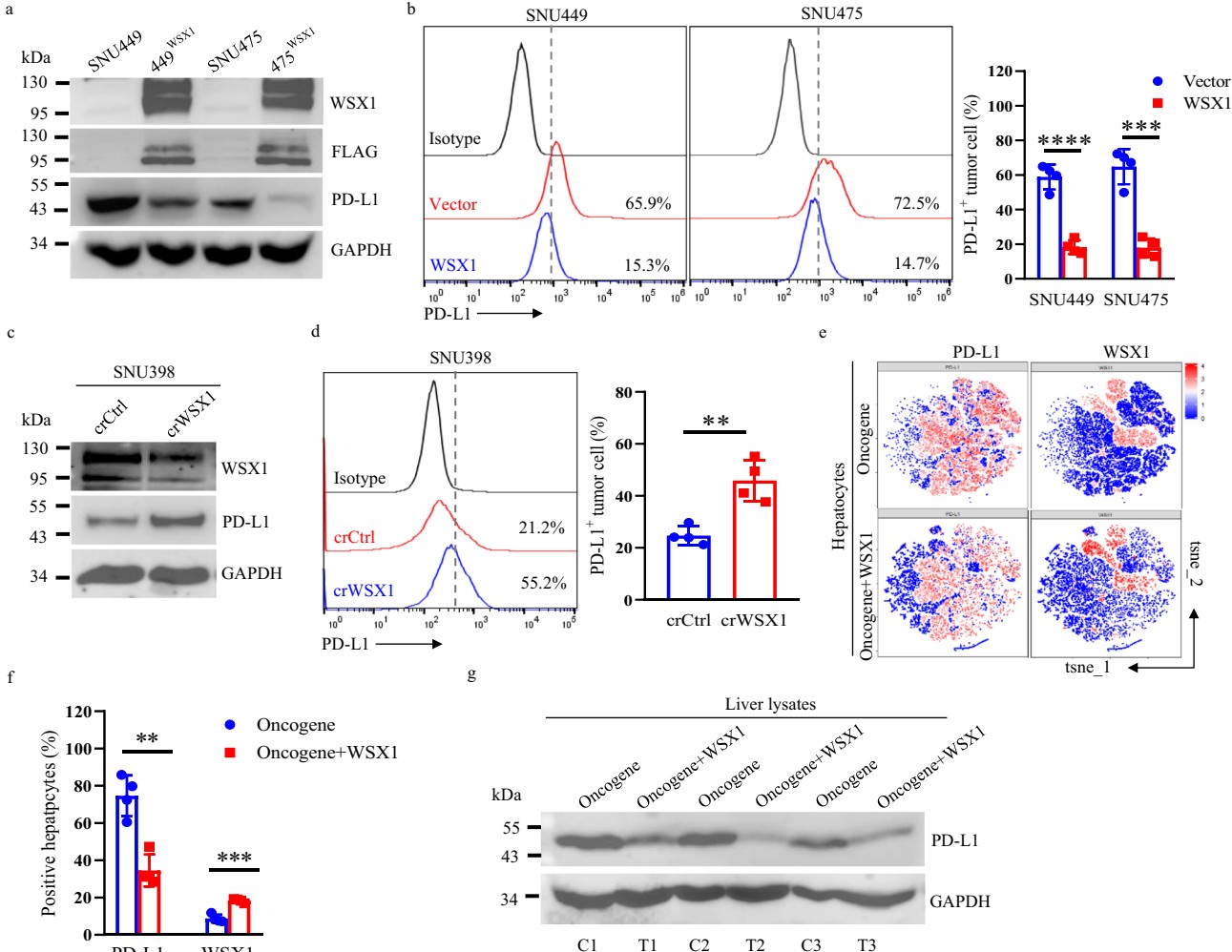

**Fig. 5 WSX1 overexpression downregulates PD-L1 expression in HCC cells. a** Protein levels of WSX1, FLAG, and PD-L1 analyzed by Western blotting. HCC cell lines SNU449 and SNU475 were transfected with plasmid DNA encoding WSX1-FLAG (449$^{WSX1}$ and 475$^{WSX1}$) or with vector plasmids. **b** Effect of WSX1 overexpression on cell surface PD-L1 expression analyzed by flow cytometry. WSX1 overexpression reduced the proportion of PD-L1$^+$ HCC cells in both SNU449 ($n = 4$ independent experiments, $P < 0.0001$) and SNU475 cells ($P = 0.0002$). **c** Immunoblotting analysis of protein levels of WSX1 and PD-L1 in SNU398 cells. CRISPR/Cas9 guiding RNAs against human WSX1 (crWSX1) were used for WSX1 knockdown, and nontargeting crRNAs were added as control (crCtrl). **d** WSX1 knockdown increased the proportion of PD-L1$^+$ HCC cells in SNU398 cells ($n = 4$ independent experiments, $P = 0.0028$). **e** t-SNE map derived from CyTOF analysis of mouse hepatocytes obtained from the HCC mouse model in Fig. 2a. Cells were color coded by the intensity of the expression levels of PD-L1 and WSX1. **f** Difference in the proportions of PD-L1$^+$ ($n = 4$ mice, $P = 0.0012$) and WSX1$^+$ mouse hepatocytes ($P = 0.0002$) based on CyTOF analysis. Data shown are representative of 2 independent experiments. **g** Immunoblotting analysis of PD-L1 protein levels in mouse liver lysates (C1–C3 represent independent samples from "oncogene" group, T1–T3 are independent samples from "oncogene + WSX1" group). All images shown are representative of 3 independent experiments. Quantitative data are presented as mean ± SD and analyzed by two-sided Student t test. **$P < 0.01$, ***$P < 0.001$, ****$P < 0.0001$. The gating strategy for sorting PD-L1$^+$ HCC cells is shown in Supplementary Fig. 8b. Source data are provided as a Source Data file.

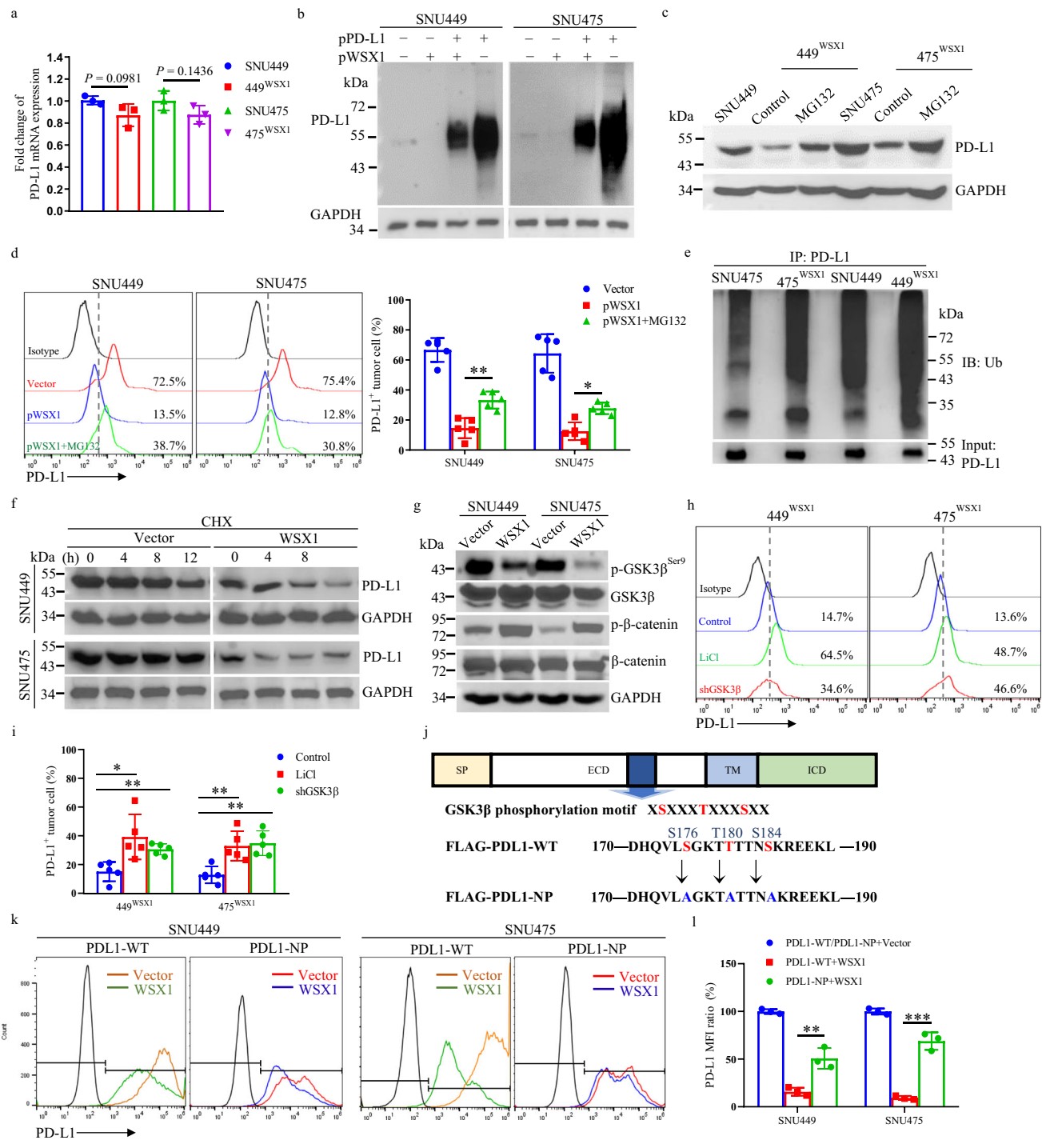

immunoblotting analysis of mouse liver lysates (Fig. 5g). Next, to further support our notion that WSX1-mediated PD-L1 down-regulation could reduce T-cell exhaustion and enhance the T cell-mediated killing effect, we performed a co-culture assay with different ratios of activated human effector T cells (E) and tumor cells (T). The survival rates for WSX1-overexpressing 449^WSX1 (E:T = 1:1, $P = 0.0029$; E:T = 2:1, $P = 0.0108$) and 475^WSX1 (E:T = 1:1, $P = 0.0022$; E:T = 2:1, $P = 0.0038$) cells were less than half of the survival rates for parental cells (Supplementary Fig. 6a). In addition, T cells that were co-cultured with 449^WSX1 ($P = 0.0131$) and 475^WSX1 ($P = 0.0017$) cells had much lower expression of PD-1 (Supplementary Fig. 6b, c). Collectively, the above results verified that WSX1 can downregulate PD-L1 expression in hepatocytes and

simultaneously reduce PD-L1/PD-1 axis-mediated T-cell exhaustion, ensuring the biological function of immune surveillance.

### WSX1 destabilizes PD-L1 by enhancing GSK3β-mediated PD-L1 protein degradation.
Motivated by our observation—down-regulation of PD-L1 by WSX1—we sought to explore the underlying molecular mechanism. To this end, we first tested whether WSX1 participated in the transcriptional regulation of PD-L1. Interestingly, WSX1 had no significant effect on PD-L1 mRNA levels in either SNU449 or SNU475 cells (Fig. 6a), implying that the regulation of PD-L1 by WSX1 is at the protein level. To validate this notion, we transfected HCC cells with

**Fig. 6 WSX1 destabilizes PD-L1 by boosting GSK3β-mediated PD-L1 degradation. a** qRT-PCR assay revealed the influence of WSX1 on PD-L1 mRNA expression level ($n = 3$ independent experiments). **b** Influence of WSX1 on the expression of exogenous PD-L1-FLAG. SNU449 and SNU475 cells were transfected with plasmid DNA encoding PD-L1-FLAG with or without WSX1 co-transfection and analyzed by Western blotting. **c** Immunoblot analysis of PD-L1 protein levels after treatment with proteasome inhibitor MG132 for 12 h. **d** Effect of MG132 on cell surface PD-L1 expression examined by flow cytometry. MG132 inhibited WSX1-mediated PD-L1 reduction in both SNU449 ($n = 5$ independent experiments, $P = 0.0027$) and SNU475 cells ($P = 0.0362$). **e** Influence of WSX1 on PD-L1 ubiquitination. Cells were pretreated with MG132 for 12 h, and cellular PD-L1 protein was pulled down by specific PD-L1 antibodies. PD-L1 ubiquitination was analyzed by anti-ubiquitin antibodies. **f** Impact of WSX1 on half-life of PD-L1 protein in HCC cells. HCC cells were treated with 25 mM CHX for 0, 4, 8, and 12 h, and cell lysates were collected separately and analyzed for PD-L1 protein levels. **g** Influence of WSX1 on protein levels of p-GSK3β$^{Ser9}$, total GSK3β, p-β-catenin$^{Ser33/Ser37/Thr41}$, and total β-catenin. β-catenin was directly phosphorylated by GSK3β at Ser33/Ser37/Thr41, and the level of p-β-catenin$^{Ser33/Ser37/Thr41}$ indirectly reflects GSK3β activity. **h** Influence of GSK3β inhibitor LiCl or GSK3β knockdown on PD-L1 expression. Cell surface PD-L1 expression on WSX1-overexpressing 449$^{WSX1}$ and 475$^{WSX1}$ cells was analyzed by flow cytometry after treatment of LiCl or transfection of GSK3β shRNAs for 48 h. **i** Statistical analysis results showing that both LiCl treatment ($P = 0.0135$ in 449$^{WSX1}$ and $P = 0.0052$ in 475$^{WSX1}$) and GSK3β knockdown ($P = 0.0020$ in 449$^{WSX1}$ and $P = 0.0015$ in 475$^{WSX1}$) increased the proportion of PD-L1$^+$ HCC cells ($n = 5$ independent experiments). **j** Schematic of site-directed mutation of the consensus motif on PD-L1-NP (S176A, T180A, and S184A), which could not be phosphorylated by GSK3β. **k** SNU449 and SNU475 cells transfected with PD-L1-wild type (WT) or PD-L1-NP (mutated) alone or co-transfected with WSX1. Alterations in cell surface PD-L1 expression levels were determined by flow cytometry. **l** Quantification of PD-L1 MFI ratio. Site-directed mutation in PD-L1-NP impaired WSX1-mediated PD-L1 downregulation in both SNU449 ($n = 3$ independent experiments, $P = 0.0066$ compared to PD-L1-WT) and SNU475 cells ($P = 0.0004$ compared to PD-L1-WT). Quantitative data are presented as mean ± SD and analyzed by one-way ANOVA or Student $t$ test. Tukey-Kramer multiple comparison test was used for pairwise comparisons in the ANOVA analysis. Unless otherwise noted, data and images shown are representative of 3 independent experiments. All statistical tests were two-sided. *$P < 0.05$, **$P < 0.01$, ***$P < 0.001$. CHX cycloheximide, SP signal peptide, TM transmembrane domain, ECD extracellular domain, ICD intracellular domain, MFI mean fluorescence intensity. The gating strategy for sorting PD-L1$^+$ HCC cells is shown in Supplementary Fig. 8b. Source data are provided as a Source Data file.

plasmids encoding FLAG-tagged PD-L1, which was not regulated by the endogenous PD-L1 promoter. As expected, WSX1 was capable of downregulating exogenous PD-L1 (Fig. 6b). Our notion was further confirmed by addition of the proteasome inhibitor MG132, which significantly restored WSX1-mediated PD-L1 downregulation in both WSX1-overexpressing cell lines, 449$^{WSX1}$ ($P = 0.0027$) and 475$^{WSX1}$ ($P = 0.0362$, Fig. 6c, d). Previous studies reported that E3 ubiquitin ligase-mediated lysine 48 (K48) polyubiquitination and subsequent proteasomal degradation control turnover of multiple proteins, including PD-L1[36]. Consistently, increased K48 ubiquitination of PD-L1 was detected in WSX1-overexpressing cells in the presence of MG132 (Fig. 6e). Moreover, the results of a pulse-chase assay revealed that WSX1 markedly reduced the half-life of the PD-L1 protein (Fig. 6f), supporting our hypothesis that WSX1 destabilizes PD-L1 through promoting its ubiquitin-mediated proteasomal degradation.

Glycogen synthase kinase 3β (GSK3β) is an essential kinase that facilitates PD-L1 phosphorylation and subsequent K48 ubiquitination[37]. Considering that WSX1 promoted PD-L1 ubiquitination, we investigated the connection between WSX1 and GSK3β in terms of PD-L1 degradation. In support of this connection, a previous report[38] found that both HCC cell lines SNU449 and SNU475, with low endogenous levels of WSX1, had an elevated basal level of phosphorylated GSK3β (p-GSK3β$^{ser9}$), representing inhibition of GSK3β kinase activity. In our study, WSX1 overexpression in SNU449 and SNU475 cells resulted in a dramatic decrease of p-GSK3β$^{ser9}$ without a significant effect on total GSK3β levels (Fig. 6g), suggesting an increase in GSK3β activity. GSK3β was reported to specifically phosphorylate β-catenin at Ser33/Ser37/Thr41 (p-β-catenin$^{Ser33/Ser37/Thr41}$)[39], whose level indirectly reflects GSK3β enzymatic activity. Notably, an increased amount of p-β-catenin$^{Ser33/Ser37/Thr41}$ was found as well, supporting our notion that WSX1 enhances GSK3β activity (Fig. 6g). It was reported that the GSK3β inhibitor LiCl prevented PD-L1 degradation[40]. In our study, treatment with LiCl substantially rescued WSX1-induced PD-L1 downregulation in 449$^{WSX1}$ ($P = 0.0135$) and 475$^{WSX1}$ cells ($P = 0.0052$, Fig. 6h, i). To further support our hypothesis, short hairpin RNAs against human GSK3β (shGSK3β) were transfected to establish WSX1-knockdown HCC cells. As a result, GSK3β knockdown substantially reversed WSX1-induced PD-L1 downregulation in

449$^{WSX1}$ ($P = 0.0020$) and 475$^{WSX1}$ cells ($P = 0.0015$, Fig. 6h, i). Furthermore, we synthesized a FLAG-PD-L1-NP expression construct from a wild-type PD-L1 construct (FLAG-PD-L1-WT) by mutating 3 phosphorylation consensus motifs on PD-L1 (S176A, T180A, and S184A; Fig. 6j), which was previously reported to completely abolish GSK3β-mediated PD-L1 phosphorylation[37]. Consistently, WSX1 prominently downregulated FLAG-PD-L1-WT but only mildly decreased FLAG-PD-L1-NP expression (SNU449: $P = 0.0066$; SNU475: $P = 0.0004$; Fig. 6k, l). These results illustrated that WSX1 downregulated PD-L1 through facilitating GSK3β-mediated PD-L1 phosphorylation and subsequent degradation.

**WSX1 enhances GSK3β activity by inactivating the PI3Kδ/AKT signaling pathway.** Protein kinase B (PKB/AKT), a serine/threonine kinase, is reported to phosphorylate GSK3β in vitro and in vivo[41,42]. Furthermore, accumulating evidence has proved that the AKT pathway is critically involved in both hepatocarcinogenesis and PD-L1 regulation[2,43]. We speculated that WSX1 enhances GSK3β activity via regulating AKT. Indeed, we found that WSX1 remarkably inhibited AKT activation, with a decrease of p-AKT$^{ser473}$ (Fig. 7a). Tuberin (TSC2) is directly phosphorylated at Thr1462 by activated AKT[44]. The suppression of AKT activity by WSX1 was further validated by the reduction of p-TSC2$^{Thr1462}$ (Fig. 7a).

The results above allowed us to formulate a pathway, in which WSX1 boosts GSK3β-mediated PD-L1 degradation through inactivating AKT. To support our notion, we transfected plasmids encoding myristoylated AKT (myr-AKT) to reinvigorate AKT activity in WSX1-overexpressing HCC cells. Indeed, transfection of myr-AKT increased both total AKT and p-AKT$^{ser473}$ protein levels (Fig. 7c). More importantly, reinvigoration of AKT activity almost completely reversed WSX1-induced PD-L1 reduction in both 449$^{WSX1}$ ($P < 0.0001$) and 475$^{WSX1}$ cells ($P < 0.000$, Fig. 7b, c). In support of this in vitro observation, our in vivo study in the HCC mouse model found a negative correlation of WSX1 expression with PD-L1 levels ($r = -0.7802$, $P = 0.0224$) and AKT activation ($r = -0.8662$, $P = 0.0054$), as measured by the protein levels of WSX1, PD-L1, and p-AKT$^{ser473}$ (Fig. 7d). Together, our results revealed a negative regulatory mechanism involved in the homeostatic control of PD-L1, in which WSX1 prevents

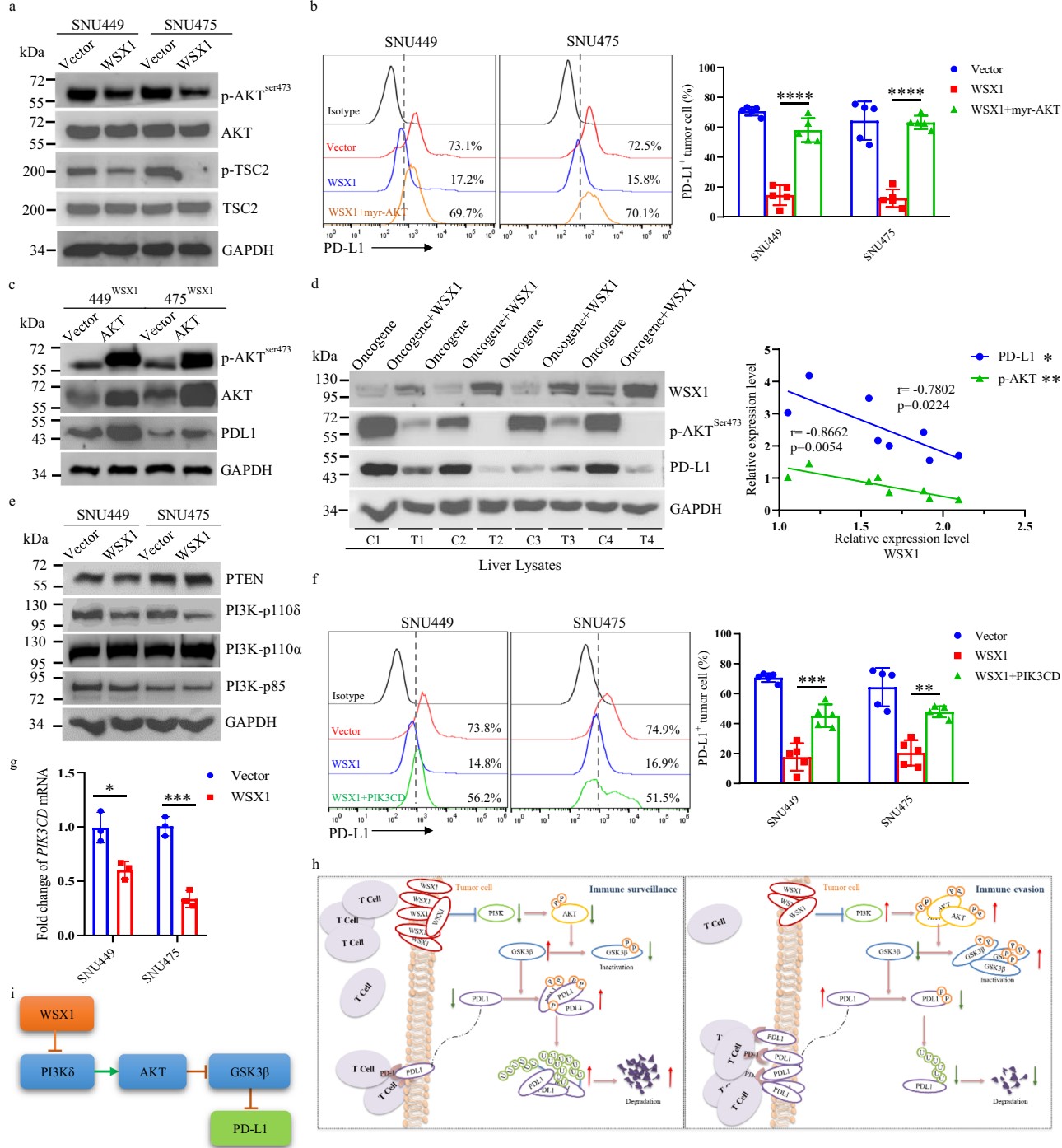

AKT-mediated GSK3β inhibition and thereby activates GSK3β for PD-L1 degradation.

Next, we investigated how WSX1 affects AKT phosphorylation. As reported, the canonical pathway leading to AKT activation is predominantly initiated by activation of phosphatidylinositol 3-kinase (PI3K), whose activity can be inhibited by the tumor suppressor PTEN[41]. PI3K encompasses 4 isoforms of the catalytic subunit, known as p110-α, -β, -γ, and -δ[45]. Among them, PI3Kδ, one of the PI3K isoforms that constitutively activates the AKT signaling pathway, was recently reported to be highly expressed in HCC and to play significant roles in malignant liver tumors[46,47]. Interestingly, WSX1 had no significant effect on PTEN, PI3K-p85 (PI3K regulatory subunit), or PI3K-p110α (PI3Kα), while remarkably downregulating PI3K-p110δ (PI3Kδ, Fig. 7e). To reveal whether the decrease of

PI3Kδ was responsible for WSX1-induced PD-L1 downregulation, we reintroduced *PIK3CD*, which encodes PI3Kδ, in WSX1-overexpressing HCC cells. Our results revealed that, as with the exogenous transfection of myr-AKT, reintroduction of PI3Kδ largely impaired the WSX1-mediated PD-L1 reduction (SNU449: $P = 0.0001$; SNU475: $P = 0.0013$; Fig. 7f). To further explore how WSX1 downregulates PI3Kδ, we performed RNA-seq and qRT-PCR analysis in both WSX1-overexpressing and parental HCC cells. Our results consistently showed that WSX1 significantly reduced *PIK3CD* mRNA levels in both 449$^{WSX1}$ ($P = 0.0139$) and 475$^{WSX1}$ cells ($P = 0.0006$), suggesting that WSX1 transcriptionally inhibits the expression of PI3Kδ (Fig. 7g). Consistently, the effect of WSX1 on PI3Kδ/AKT/GSK3β/PD-L1 signaling was also confirmed in WSX1-knockdown cells (Supplementary Fig. 7).

**Fig. 7 WSX1 relieves inhibition of GSK3β activity through impeding the PI3Kδ/AKT signaling pathway. a** Effect of WSX1 overexpression on protein levels of p-AKT$^{Ser473}$, AKT, p-TSC2$^{Thr1462}$, and TSC2 in HCC cells determined by immunoblotting. TSC2 is directly phosphorylated at Thr1462 by AKT; thus p-TSC2$^{Thr1462}$ levels indirectly reflect AKT activity. **b** Impact of AKT reactivation on WSX1-mediated PD-L1 downregulation detected by flow cytometry. SNU449 and SNU475 cells were transfected with WSX1 alone or co-transfected with membrane-bound myr-AKT. Statistical analysis results are shown on the right ($n = 5$ independent experiments, $P < 0.0001$ in both SNU449 and SNU475 cells). **c** Impact of myr-AKT transfection on protein levels of p-AKT$^{Ser473}$, total AKT, and PD-L1 in 449$^{WSX1}$ and 475$^{WSX1}$ cells. **d** Immunoblotting analysis of expression of WSX1, PD-L1, and p-AKT$^{Ser473}$ in whole liver lysates obtained from FVB/NJ mice injected with *NRAS/AKT* oncogenes with or without WSX1 (left, C1–C4 represent independent samples from "oncogene" group, T1–T4 are independent samples from "oncogene + WSX1" group). Statistical analysis of correlations of WSX1 with PD-L1 and p-AKT$^{Ser473}$ expression in mouse livers (right). **e** Effect of WSX1 on expression of PTEN, PI3K-p85, PI3K-p110α, and PI3K-p110δ. **f** Impact of *PIK3CD* overexpression on WSX1-mediated PD-L1 reduction (left). *PIK3CD* encodes PI3K-p110δ, which is the key component of PI3Kδ. Reintroduction of *PIK3CD* impaired WSX1-induced PD-L1 reduction in both SNU449 ($n = 5$ independent experiments, $P = 0.0001$) and SNU475 cells ($P = 0.0013$). **g** Effect of WSX1 on *PIK3CD* mRNA levels in SNU449 ($n = 3$ independent experiments $P = 0.0139$) and SNU475 HCC cells ($P = 0.0006$). **h** A proposed model illustrating that WSX1 facilitates antitumor immune surveillance through inhibiting the PI3Kδ/AKT/GSK3β/PD-L1 signaling pathway. Under physiological conditions, highly expressed WSX1 in hepatocytes transcriptionally downregulates PIK3δ, thereby reducing AKT activation and subsequently liberating GSK3β kinase activity from inhibition by AKT, leading to boosted GSK3β-mediated PD-L1 degradation. Without excessive PD-L1 expression on tumor cells, effector CD8$^+$ T cells maximize their killing effect, resulting in inhibition of HCC development (Left). Otherwise, lack of WSX1 results in uncontrolled neoplastic PD-L1 expression and, ultimately, tumor immune evasion (Right). **i** Schematic diagram of the interactions among WSX1, PI3Kδ, AKT, GSK3β, and PD-L1. Quantitative data are presented as mean ± SD and were analyzed by one-way ANOVA or Student *t* test. Tukey-Kramer multiple comparison test was used for pairwise comparisons in the ANOVA analysis. Correlation analyses were performed by Pearson correlation test. Unless otherwise noted, data and images shown are representative of 3 independent experiments. All statistical tests were two-sided. *$P < 0.05$, **$P < 0.01$, ***$P < 0.001$. myr-AKT myristoylated AKT. The gating strategy for sorting PD-L1$^+$ HCC cells is shown in supplementary Fig. 8b. Source data are provided as a Source Data file.

In summary, on the basis of the above results, we are able to reveal an interaction model of WSX1, PI3Kδ, AKT, GSK3β, and PD-L1 (Fig. 7h, i). Specifically, in physiological conditions, WSX1 is highly expressed in hepatocytes and contributes to the tight control of PD-L1 levels through governing the PI3Kδ/AKT/GSK3β signaling pathway, ensuring a homeostatic PD-L1 expression level on normal hepatocytes, and thereby enabling effective CD8$^+$ T-cell-mediated immunosurveillance. However, when WSX1 is downregulated by multiple oncogenic signals, PI3Kδ escapes from WSX1-mediated transcriptional inactivation, resulting in the subsequent activation of AKT. Hyperactive AKT then inactivates GSK3β and thereby blocks GSK3β-mediated PD-L1 degradation, which results in excessive PD-L1 expression on malignant hepatocytes, leading to PD-L1/PD-1 axis-mediated tumor immune evasion and, ultimately, HCC development.

## Discussion

Over the past few decades, tumor suppressor genes have been thoroughly investigated for their indispensable role in maintaining genetic integrity in a cell-autonomous manner. These genes can be broadly classified into 2 classes: "gatekeepers," which regulate the cell cycle and replication, and "caretakers," which preserve genetic stability[48,49]. More recently, as more and more researchers have realized the crucial role of the immune system in tumor development[5,50], ample evidence has suggested that there might be a third class of tumor suppressor gene, "guardians of immune integrity," which ensure effective immunosurveillance[51–54]. Several classical tumor suppressor genes, such as *TP53*, *PTEN*, and *RB1*, have been recently implicated in tumor immunology as a functional extension of their conventional cell-autonomous tumor-suppressive roles. However, none of their tumor-suppressive functions exclusively depend on immunosurveillance[51–54]. In contrast to these dual-function tumor suppressor genes, WSX1's tumor suppressor function exclusively relies on its regulation of adaptive CD8$^+$ T-cell immunosurveillance. Neither tumor-cell proliferation (gatekeeper role) nor genetic stability (caretaker role) was significantly affected by WSX1. Instead, WSX1 acts as an "immune surveillance defender," blocking oncogene-induced HCC tumorigenesis in a non-cell-autonomous manner. WSX1 restrained neoplastic PD-L1 expression and protected CD8$^+$ T cell-mediated

antitumor immunosurveillance from exhaustion, thus ensuring effective immune elimination of malignant cells.

T-cell exhaustion has a major role in immune dysfunction and tumor immune evasion[29]. Among all tumor-infiltrating lymphocytes, CD8$^+$ T cells are the main subset that performs antitumor immunity through executing T-cell receptor-mediated killing of malignant cells[55]. The correlation of CD8$^+$ T-cell infiltration with improved overall survival has been well established[55,56]. Unfortunately, tumor-infiltrating CD8$^+$ T cells are frequently in a state of exhaustion, characterized by a progressive loss of effector functions, robust activation of the T-cell exhaustion driver TOX[31–34], and sustained expression of inhibitory receptors such as PD-1, LAG-3, Tim-3, and CTLA-4[28,29]. Indeed, our results demonstrated severe CD8$^+$ T-cell exhaustion in the *NRAS/AKT* oncogene-driven spontaneous HCC mouse model. However, WSX1 expression in livers reduced CD8$^+$ T-cell dysfunction, evidenced by upregulation of functional markers and downregulation of multiple inhibitory receptors and TOX. Intriguingly, either the use of immune-deficient mice or in vivo depletion of CD8$^+$ T cells completely reversed the tumor-suppressive effect of WSX1, indicating that CD8$^+$ T-cell immunity is indispensable for WSX1-induced HCC suppression. Overall, instead of directly eliminating malignant cells, WSX1 prevented T-cell exhaustion and thus maximized the activity of cytotoxic CD8$^+$ T cells.

Although T-cell exhaustion has been demonstrated in a variety of human cancers[29], the underlying mechanisms contributing to its development remain poorly understood. Currently, the intrinsic negative regulatory signaling mediated by inhibitory receptors and the extrinsic suppressive tumor microenvironment are generally accepted as the pivotal pathways driving T-cell exhaustion[29]. In our study, WSX1 injection resulted in a remarkable upregulation of WSX1 on malignant hepatocytes without impacting its expression on infiltrating CD8$^+$ T cells, suggesting that the regulation of T-cell exhaustion by WSX1 is indirect and most likely due to modification of malignant cells. Considerable evidence supports the idea that engagement of PD-L1 on cancer cells with its receptor, PD-1, on effector T cells is the major mechanism contributing to the exhaustion of tumor-infiltrating lymphocytes and subsequent tumor immune evasion[9,29]. Indeed, our investigation revealed that WSX1 significantly downregulated PD-L1 on HCC cells and

enhanced T cell-mediated tumor eradication. In addition to cell surface inhibitory receptors, several immunoregulatory cytokines have been linked to T-cell exhaustion[28,29]. Immunosuppressive cytokines such as IL-10 and TGF-β play a positive role, while γ-chain cytokines, including IL-2, IL-7, and IL-21, have been implicated in antagonizing T-cell dysfunction[28,29]. A more recent study reported that cancer cell-derived cholesterol devitalized T cells by modulating endoplasmic reticulum stress pathways, highlighting the importance of metabolic factors in immune regulation[57]. Based on our RNA-seq data, no significant alteration of the genes encoding the above immunoregulatory cytokines or metabolites was detected, but their involvement cannot be excluded yet and needs further investigation.

Current immunotherapies targeting the PD-L1/PD-1 axis have exhibited promising clinical responses in multiple tumor types[58]. However, their benefit for overall survival is not satisfactory owing to intrinsic or acquired resistance[59]. Additionally, PD-1 and PD-L1 are located not only on tumor cells but also on normal cells; therefore, nonselective blockade of the PD-L1/PD-1 interaction inevitably causes unfavorable effects on immune homeostasis[43]. In this case, understanding the physiological balance system controlling PD-L1 would benefit the development of more effective therapeutic strategies. Over the past decades, numerous factors, such as IFN-γ, lactate, and NF-κB, have been found to upregulate PD-L1 expression at translational or post-translational levels[43]. Considering that the immune system is a tightly regulated network that maintains a delicate and finely tuned balance between immunity and tolerance, there should be a matched control system to restrain immune checkpoints. However, few negative regulators of PD-L1 have been identified. Limited studies claimed that some microRNAs, such as miR-513, could inhibit PD-L1 translation[60]. Recently, a study reported that cyclin D-CDK4 kinase destabilized PD-L1 through upregulating cullin 3-SPOP E3 ligase, which was involved in ubiquitination-mediated PD-L1 degradation[61]. Even so, little is known about the negative control of PD-L1 protein turnover. In this study, we discovered that WSX1 serves as a negative force actively participating in the homeostatic control of PD-L1.

Accumulating evidence demonstrates that PD-L1 is extensively regulated by the ubiquitin/proteasome system[43,62,63]. GSK3β, a constitutively activated serine/threonine kinase, was reported to mediate PD-L1 phosphorylation, which facilitated ubiquitin E3 ligase recognition and subsequent ubiquitination of PD-L1[37]. Activated AKT deactivates GSK3β[42,64]. In our study, we found that WSX1 destabilized PD-L1 protein via downregulating AKT and prompting GSK3β-mediated ubiquitination and subsequent proteasomal degradation. Reintroduction of myr-AKT almost completely reversed WSX1-induced PD-L1 reduction. We also noted that either GSK3β knockdown or GSK3β inhibitor LiCl only partially impaired the WSX1-induced effect, indicating that other pathways downstream of AKT might be involved.

Substantial research has shown that AKT is activated in a PI3K-dependent manner[41,65]. PI3Kδ, one of the PI3K isoforms, converts phosphatidylinositol (3,4)-bisphosphate (PIP2) lipids to phosphatidylinositol (3,4,5)-trisphosphate (PIP3) lipids, which in turn bind to AKT and allow its activation[45,66,67]. PI3Kδ was previously recognized to be typically expressed in cells of hematopoietic origin[66]. However, increasing evidence proved that PI3Kδ was also highly expressed in several solid tumors. PI3Kδ was reported to promote breast and prostate cancer cell proliferation via dampening of PTEN activity and to induce colorectal cancer cell growth and invasion by activating the AKT/GSK3β/β-catenin signaling pathway[68,69]. Recently, highly expressed PI3Kδ was also found in HCC and closely correlated with poor survival rates[46]. Interestingly, our results revealed that WSX1 reduced both PI3Kδ mRNA and protein levels and that

PI3Kδ reintroduction largely counteracted WSX1-induced PD-L1 downregulation. Taken together, we reveal a signaling pathway in which WSX1 transcriptionally downregulates PI3Kδ without impacting other classical isoforms of PI3Ks and thereby reduces AKT activity, which in turn prevents AKT-mediated GSK3β inhibition, leading to increased GSK3β-mediated PD-L1 degradation. However, as no study has reported a role for WSX1 as a transcription factor, the mechanism underlying the transcriptional inhibitory effect of WSX1 on PI3Kδ needs further investigation.

Notably, chemical AKT inhibitors were reported to inhibit tumor-intrinsic phenotypes and PD-L1 transcription[70,71], but WSX1 has no significant effect on either PD-L1 mRNA or HCC cell proliferation and migration, indicating a unique regulatory characteristic of WSX1 on AKT/GSK3β/PD-L1 signaling that cannot be mimicked by traditional chemical inhibitors. Isoform-specific effects of the individual PI3Ks on AKT activity might be responsible for this difference, at least in part, but the underlying mechanism remains unclear and needs further exploration.

Moreover, in addition to the ubiquitination-mediated proteasomal degradation of PD-L1, it has been reported that glycosylated modification, subcellular transportation, and lysosomal degradation are also closely associated with PD-L1 protein turnover[43,72]. For this reason, further study is required to uncover the overall picture of the functional mechanisms of WSX1.

Collectively, our studies revealed a tumor suppressor gene, *WSX1*, which functions as a "guardian" of CD8+ T cell-mediated cancer immunosurveillance and acts as a homeostatic "supervisor" of the immune checkpoint PD-L1. Considering that evidence has illustrated the indispensable role of the liver in host immune homeostasis[5]—and given the high expression of WSX1 in normal liver tissues and its close interaction with both the critical oncogenic PI3K/AKT signaling pathway and the immune checkpoint PD-L1—the biological function of WSX1 we have uncovered here might only be the tip of the iceberg; the importance of WSX1 might be greater than we thought. Notably, WSX1 is expressed in multiple normal organs besides the liver, including the colon, intestine, and kidney, implying a broader role for WSX1 in immune homeostasis, which might yield insights into the development of more effective immunotherapies.

## Methods

**Human tissue microarrays.** Our study was approved by the Institutional Review Board of The University of Texas MD Anderson Cancer Center (#PA12-0604). Human tissue microarrays (FDA662a, BC03116a, and HLiv-HCC180Sur-03) were purchased from Biomax in accordance with a protocol approved by MD Anderson's Institutional Review Board. FDA662a contains 33 types of normal human organ tissue (cerebrum, cerebellum, peripheral nerve, adrenal gland, thyroid gland, spleen, thymus gland, bone marrow, lymph node, tonsil, pancreas, liver, esophagus, stomach, small intestine, colon, lung, salivary gland, larynx, kidney, bladder, testis, prostate, penis, ovary, fallopian tube, breast, endometrium, cervix, cardiac muscle, skeletal muscle, mesothelium, and skin), with samples from each organ taken from 2 individuals. BC03116a contains samples from 40 cases of HCC, 17 normal liver tissues, and 13 NAT. HLiv-HCC180Sur-03 contains samples from 90 cases of HCC and paired NAT with survival follow-up information for 2.0 to 3.7 years. All tissue specimens were obtained from surgical patients of Asian descent from January 2010 to September 2011. No case of overlap was found between BC03116a and HLiv-HCC180Sur-03.

**Spontaneous HCC mouse model.** Eight- to 10-weeks-old C57BL/6J, FVB/NJ, and NSG mice were purchased from the Jackson Laboratory. WSX1⁻/⁻ mice with a C57BL/6J background were previously donated by Dr. Frederic de Sauvage (Genentech). IL-27p28⁻/⁻ mice were generated as previously described[73]. All mice were maintained and treated in accordance with ethical guidelines approved by the Institutional Animal Care and Use Committee (IACUC) at MD Anderson.

Because simultaneous activation of AKT/mTOR and RAS/MAPK pathways is often found in human HCC[25], previous studies established a spontaneous HCC mouse model based on HDI with plasmids encoding NRASV12/myr-AKT, combined with a transposon system to deliver oncogenes. As previously described[25,26], 20 μg of pT3-myr-AKT-HA and pT/Caggs-NRASV12 mixed with sleeping beauty transposase pCMV(CAT)T7-SB100 in a ratio of 25:1 were diluted

in 2 mL filtered Dulbecco phosphate buffered saline (DPBS) and then delivered via HDI into the lateral tail vein of mice in 5–7 s at day 0. PiggyBac Dual promoter (PB513B-1) was included as a negative control. Next, a vector plasmid or plasmid encoding WSX1 diluted in 2 mL DPBS was delivered via HDI weekly. The mice were monitored daily. Due to the aggressive tumor growth, all mice died or were humanely euthanized within 8 weeks after the first injection.

**Antibodies and reagents**. The following antibodies were used: primary antibody against PD-L1 (Proteintech 17952 and 66248, GeneTex GTX31308), pan-AKT (Cell Signaling Technology 4685), phosphor-AKT$^{S473}$ (Abcam ab81283, Cell Signaling Technology 9271), GSK3β (Cell Signaling Technology 12456), phosphor-GSK3β$^{Ser9}$ (Cell Signaling Technology 5558), WSX1 (Thermo Fisher PA5-96963), β-catenin (Cell Signaling Technology 8480), phosphor-β-catenin$^{Ser33/37/Thr41}$ (Cell Signaling Technology 9561), PTEN (Cell Signaling Technology 9188), phospho-TSC2$^{Thr1462}$ (Cell Signaling Technology 3617), TSC2 (Cell Signaling Technology 4308), PI3K-p85 (Cell Signaling Technology 4292), PI3K-p110α (Cell Signaling Technology 4255), PI3K-p110δ (Cell Signaling Technology 34050), FLAG (Cell Signaling Technology 2368), Ub (Santa Cruz Biotechnology sc-8017), and WSX1 (clone 237, Monoclonal Antibodies Core Facility at MD Anderson Cancer Center)[74]. PerCP/Cyanine5.5 anti-mouse CD3 (BioLegend 100328), V450 anti-mouse CD8a (Tonbo 75-0081), FITC anti-mouse CD4 (BioLegend 100405), V450 anti-mouse NK1.1 (BD biosciences 560524), PE/Cy7 anti-mouse PD-1 (BioLegend 109109), PE anti-mouse CTLA-4 (BioLegend 106306), PE/Cy7 anti-mouse LAG-3 (BioLegend 125225), PE anti-mouse Tim3 (BioLegend 134009), PE anti-mouse granzyme B (ebioscience 12-8898-80), PE anti-mouse Ki67 (BioLegend 652404), PE anti-mouse perforin (ebioscience 12-9392-82), PE anti-human CD3 (BioLegend 300308), PE/Cy7 anti-human PD-1 (BioLegend 367414), APC anti-mouse TOX (Miltenyi Biotec 130-118-335), APC anti-mouse IFN-γ (ebioscience 17-7311-82), PE anti-mouse IL-2 (BioLegend 503808), PE anti-human WSX1 (R&D FAB14791P), and HRP anti-human/mouse GAPDH (Proteintech HRP-6000). Anti-mouse CD8α (clone 2.43), anti-mouse CD4 (clone GK1.5), and anti-mouse NK1.1 (clone PK136) for immune cell depletion were constructed and purchased from BioXCell. LiCl, cycloheximide and MG132 were obtained from Sigma-Aldrich. The dilution of each antibody is shown in Supplementary Table 1.

**Cell culture, plasmids, and transfection**. Human HCC cell lines SNU398, SNU449, SNU475, HepG2, and Hep3B were purchased from ATCC and were independently validated using short tandem repeat DNA fingerprinting at MD Anderson. Human primary T cells were obtained from Lonza (3W-350). Cells were grown in RPMI 1640 medium supplemented with 10% fetal bovine serum and 1% antibiotic mixture.

pcDNA3.1-WSX1-FLAG, pcDNA3.1-WSX1-eGFP, and pcDNA3.1-PD-L1-FLAG plasmids were obtained from GenScript. The PD-L1-NP mutant (S176A/T180A/S184) plasmid was generated via site-directed mutagenesis of the pcDNA3.1-PD-L1-FLAG expression vector from GenScript. pT3-myr-AKT-HA was a gift from Xin Chen (Addgene plasmid 31789)[75]. pT/Caggs-NRASV12 was a gift from John Ohlfest (Addgene plasmid 20205)[76]. pCMV(CAT)T7-SB100 was a gift from Zsuzsanna Izsvak (Addgene plasmid 34879)[77]. pDONR223-PIK3CD-WT was a gift from Jesse Boehm, William Hahn, and David Root (Addgene plasmid 82222)[78]. PiggyBac Dual promoter (PB513B-1) was purchased from System Bioscience. All constructs were confirmed using enzyme digestion and DNA sequencing. All plasmids were purified using the GenElute Endotoxin-Free Plasmid Maxiprep Kit (Sigma) before injecting into mice.

WSX1-stable transfectants in SNU449 and SNU475 cells were generated using a retroviral-based GFP-WSX1 expression system and were selected by FACSAria Cell Sorter (BD Biosciences). For transient transfection, cells were transfected with plasmid DNA using jetPRIME® transfection reagent (Polyplus) or electrotransfection. For GSK3β knockdown, cells were treated with a human GSK3β shRNA expression vector as described in a previous study[79]. The GSK3β shRNA sequence was 5′-GAAAGCTAGATCACTGTAA-3'[6]. In addition, scrambled shRNAs were added as a negative control (shCtrl). A CRISPR/Cas9 approach was used for WSX1 knockdown. The WSX1 guide RNA sequences (crWSX1: CCTCACCAGAAGGCGGTGTC) were cloned into the px458 vector. The constructs were then transfected into HCC cells using jetPRIME® transfection reagent. Nontargeting crRNAs were added as control (crCtrl).

**In vivo CD4$^+$ T-cell, CD8$^+$ T-cell, and NK cell depletion**. Depletion antibodies against CD8$^+$ T cells (clone 2.43), CD4$^+$ T cells (clone GK1.5), NK cells (clone PK136), or their matched IgG isotypes were administered intraperitoneally into mice twice a week for 3 weeks at a dose of 250 μg per mouse, starting on day 3 after the first HDI of *NRAS/AKT* oncogenes. Efficiency of in vivo immune cell depletion was confirmed by flow cytometry.

**T-cell-mediated tumor cell-killing assay**. Human primary T cells were activated with 100 ng/mL CD3 antibody and 10 ng/mL IL-2, and then co-cultured with human HCC cells in 12-well plates at ratios of 1:1 or 1:2, respectively. After co-culture for 48 h, a SYTOX™ AADvanced™ Dead Cell Stain Kit was used to exclude dead cells. Next, the mixed cells were stained with PE-conjugated anti-human

CD3 specific antibodies to distinguish live T-cells and live HCC cells and then evaluated using a BD FACSCanto II cytometer.

**Immunohistochemical analysis**. Paraffin-embedded human tissue microarrays or mouse liver/tumor tissue sections were deparaffinized, rehydrated, subjected to heat-induced antigen retrieval, blocked in goat serum blocking buffer, and then incubated with primary antibodies overnight at 4 °C. The next day, the sections were washed and then incubated with biotin-conjugated secondary antibodies for 1 h at room temperature, and then the following detection/visualization kits were used: ABC enhanced Vectastain kit system (Vector Laboratories), DAB peroxidase substrate kit, and hematoxylin and eosin. The images were captured using a Nikon Eclipse Ti microscope and analyzed by ImageJ version 1.8.0.

**Cell profiling using CyTOF**. Single-cell suspensions of mouse liver were made as described previously[80]. Livers were purfused with cold PBS, mechanically disrupted, and and digested for 20 min at 37 °C in Gey's balanced salt solution (GBSS) with 50 μg/ml collagenase IV, then filtered through a 250-μm cell strainer. Tumor-infiltrating lymphocytes and hepatocytes were separated using a Ficoll-Paque Plus gradient centrifugation. For CyTOF assay, single T-cells or hepatocytes were incubated with a mixture of metal-labeled antibodies (Supplementary Table 2) for 30 min at room temperature and then incubated with Cell-ID Intercalator-$^{103}$Rh overnight at 4 °C. The labeled samples were detected using a CyTOF 2 mass cytometer (Fluidigm) at the Flow Cytometry and Cellular Imaging Facility at MD Anderson. CyTOF results were analyzed by R package software (R Development Core Team, Version 3.5.3).

**Western blotting, co-immunoprecipitation, and pulse-chase assay**. For Western blotting analysis, tissues or cells were subjected to lysis in RIPA buffer (50 mM Tris-HCl, pH 7.4; 1% NP-40, 0.25% sodium deoxycholate; NaCl 150 mM; EDTA 1 mM) supplemented with 50 mM NaF, 20 mM β-glycerophosphate, and a complete protease inhibitor cocktail (Roche Diagnostics, Indianapolis, IN, USA). Equal amounts of tissue or cell lysates were separated by 10% sodium dodecyl sulfate-polyacrylamide gel electrophoresis (SDS-PAGE) and transferred to nitrocellulose membranes by an iBlot gel transfer device (Invitrogen). The membranes were blotted with primary and secondary antibodies to detect the proteins of interest. GAPDH was used as a loading control. For co-immunoprecipitation, cells were lysed in lysis buffer (Tris-HCl 50 mM, pH 8.0; NaCl 150 mM; EDTA 5 mM; 0.5% NP-40). Lysates were incubated with antibodies on a rotating wheel overnight at 4 °C and then pulled down with protein A/G agarose beads (Pierce) at 4 °C for 6 h. Beads were collected by centrifugation, washed, and boiled in 2× SDS-PAGE sample buffers and analyzed by Western blotting. For pulse-chase assay, cells were treated with 25 mM cycloheximide for 0, 4, 8, or 12 h, and cell lysates were collected separately and subjected to Western blot analysis to detect PD-L1 protein.

**Flow cytometry**. Single-cell suspensions were stained with the indicated fluorescence-conjugated primary antibodies for 30 min at 4 °C and then analyzed on an Attune acoustic focusing cytometer (Applied Biosystems). For intracellular staining, cells were fixed and permeabilized before incubating with antibodies. Staining for nuclear transcription factor was performed according to a protocol provided by eBioscience. Stained cells were isolated by flow cytometry and the results were analyzed by FlowJo software. The gating strategies were shown in Supplementary Fig. 8.

**qRT-PCR assay**. Total RNA was extracted from HCC cells using a RNeasy Plus Mini Kit (QIAGEN, Venlo, Netherlands) according to the manufacturer's instructions and then subjected to complementary DNA synthesis by reverse transcription using a SuperScript III kit (Invitrogen). Quantitative reverse transcriptase PCR (qRT-PCR) assays were performed according to the manufacturer's instructions using a StepOnePlus Real-Time PCR System (Life Technologies). The primers were as follows: 5′-TCACTTGGTAATTCTGGGAGC-3′ (PD-L1 forward), 5′-CTTTGAGTTTGTATCTTGGATGCC-3′ (PD-L1 reverse), 5′-CATATGTG CTGGGCATTGGC-3′ (PI3Kδ forward), 5′-TTTCACAGTAGCCCCGGAAC-3′ (PI3Kδ reverse), 5′-GAGTCAACGGATTTGGTCGT-3′ (GAPDH forward), 5′-GACAAGCTTCCCGTTCTCAG-3′ (GAPDH reverse). All the data analyses were performed using the comparative Ct method. GAPDH was used as the internal control.

**Cell proliferation and migration assay**. Cell proliferation was analyzed by Cell Counting Kit-8 (CCK-8) assay as described[81,82]. HCC cells were plated in 96-well plates (4 × 10$^3$ cells per well) in DMEM supplemented with 10% FBS and cultured for indicated time. Next, the CCK8 reagent was added to each well according to the manufacturer's instructions. Results were expressed as the absorbance of each well at 450 nm as measured using a microplate spectrophotometer (Multiskin spectrum, Thermo Cooperation, America).

Cell migration assays were performed using Transwell chambers (Costar; Corning)[83–85]. Briefly, cells (4 × 10$^4$ cells per chamber) were seeded in serum-free medium in the upper chamber, and 10% FBS was used as chemoattractant in the bottom. Cells were cultured for 24 h. The migrated cells were stained with crystal

violet and photographed at ×200 magnification. All measurements were performed in triplicates.

**Statistical analysis**. All quantitative data are presented as mean ± standard deviation (SD). Statistical significance was determined by two-tailed Student $t$ test or one-way analysis of variance (ANOVA). Tukey-Kramer multiple comparison test was used for pairwise comparisons in the ANOVA analysis. Normality of data was assessed using the Shapiro–Wilk test of normality. Differences in survival curves were analyzed by Kaplan–Meier analysis and the log-rank test. Correlation analyses were performed by Pearson correlation test. All statistical tests were two-sided and conducted using GraphPad Prism 8 software (GraphPad Software, La Jolla, CA). Differences were considered significant at $P < 0.05$. All data shown are representative of at least 2 independent experiments.

**Reporting summary**. Further information on research design is available in the Nature Research Reporting Summary linked to this article.

## Data availability
The authors declare that all data supporting the findings of this study are available within the paper and its Supplementary Information files or from the corresponding author upon reasonable request. Source data are provided with this paper.

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

## Acknowledgements

This study is supported in part by a grant from the U.S. National Institutes of Health (R01DK102767). We are grateful for the professional support from the MDACC Research Histology Core Laboratory and North Campus Flow Cytometry and Cellular Imaging Core Facility. Editorial support was provided by Dawn Chalaire, Amy Ninetto, and Bryan Tutt in Editing Services, Research Medical Library, The University of Texas MD Anderson Cancer Center.

## Author contributions

S.L. and M.W. initiate the original idea and planned the experiments for this work. M.W., X.X. and J.H. perform the experiments. M.W. collects data and perform statistical analyses. M.W. and S.L. prepare the manuscript draft with input from all authors. N.W.F. provides histological evaluation of mice tissues. All authors provide critical feedback and help shaping the research, analysis, and manuscript.

## Competing interests

The authors declare no competing interests.
