## [Peer Review File · Nature Communications]

Reviewers' comments:

Reviewer #1 (Remarks to the Author): expertise in HCC

In this study, the authors investigated the mechanism by which tumor cells escape from host immunosurveillance and examined the role of a potential tumor suppressor gene, *WSX1*, on HCC development. Mechanistically, they showed that *WSX1* induces PD-L1 degradation via AKT/GSK3 β pathway in tumor cells, which may in turn reduce CD8+T cell functional exhaustion via PD-1/PD-L1 interaction to suppress tumor development. While these findings are of scientific values, some of the findings are not well supported by the current data.

Major comments:

1. Fig. 1a: The expression of *WSX1* was examined in a human multiple normal tissue microarray. Most of the 33 types of normal tissues showed high level of staining signal, including the liver whose signal was even higher than in immune cell-enriched tissues. How is the specificity of the antibody against *WSX1* verified? The authors must provide negative controls for the immunostaining e.g. using antigen-absorbed antibody, otherwise the authenticity of the expression pattern in normal and malignant tissues is in doubt.
2. Fig. 1c: According to the authors, there were 130 cases of HCC in the human liver tissue microarrays (BC03116a and HLiv-HCC180Sur-03). But why "HCC patients were manually divided into low (n = 47) and high (n = 43) *WSX1* expression groups"? How about the other 40 cases that were excluded from the survival analysis? Moreover, what is the expression level of *WSX1* and its relationship with patient survival in TCGA data?
3. Fig. 2b, 2f, 3k, s2b, s2e, s4b: Tumor weight, tumor volume or number of tumor nodule instead of liver weight should be shown for accurate assessment of tumorigenicity.
4. Fig. 3a and 4a: The tissues used for MS-CyTOF analysis were not clearly described. It was mentioned in the main text that 'intrahepatic infiltrating immune cells' were isolated for Fig. 3. However, in the figure legend for Fig. 4, 'tumor cells isolated from mice livers' were analyzed. As the immune cell profiles and gene expressions between tumor and adjacent liver tissues would be vastly different, the authors should examine the immune cell profiles and gene expressions in both tumor and liver tissues separately and explicitly mention the tissue identity involved in the data.
5. Fig. 3a: The authors stated that 'the majority is T cells'. How to define 'majority', by cell number or percentage or cluster numbers? Since the size of the tumor and liver tissues in the 'Oncogene' and 'Oncogene + *WSW1*' groups are very different as shown in Fig. 2a, are the differences observed in Fig. 3a due to *WSX1* expression or tissue difference? Without clear description of tissues involved, the meaning of the data is not clear.
6. Fig. 3b and 3c: Notably, *WSX1* treatment significantly increased the proportion of infiltrating T cells from 41.43% \pm 5.87% to 54.27% \pm 1.76% (P = 0.0221, n = 3, Figure 3b), while *WSX1* had no significant effect on the proportion of CD4+, CD8+, and CD4+CD8+ T cells (Figure 3c). This statement is confusing. Where were T cells infiltrating to? Tumor or liver tissues?
7. Why is the proportion of T cells increased if the key function of *WSX1* is to maintain T cell activity through down-regulating PD-L1?
8. Fig. 3d: Which tissue does the heatmap of T cell panel markers correspond to, 'Oncogene' or 'Oncogene + *WSW1*'? The heatmaps of both groups should be shown.
9. For the comprehensive characterization of T cell subtypes, gamma-delta T and NKT cells should also be included in the antibodies panel for CyTOF.
10. Fig. 3h: The authors used PD1+LAG3+CTLA4+ to define exhausted CD8+T cells, and PD1-LAG3-CTLA4- to define non-exhausted CD8+T cells. However, as one of the important markers of functional exhausted CD8+T cells, Tim3 was not included in the analysis.

11. Fig. 3i: How to explain the reduction of GranB and Ki67 in the 'Oncogene + WSX1' group when compared to the 'Oncogene' group? These activated T cell markers are not supposed to reduce together with other inhibitory markers. This data casts doubt on whether WSX1 reduce T cell exhaustion or T cell function? To clarify this, the authors should measure the levels of other functional markers such as IFN-g, TNF-a, and perforin to define the functional status of T cells in different groups.

12. Fig. 3: More importantly, the CyTOF data needs to be independently validated by multi-color flow cytometry using specific subsets of exhaustion and functional markers. This will greatly help to resolve the confusing points as listed above.

13. Fig. 3j-L: The data of the various immune cell depletion was impressive. Yet, the levels of immune cells (CD8T, CD4T and NK) should be determined to confirm the depletion. The authors concluded that 'Intriguingly, either adopting immune-deficient mice or in vivo depletion of CD8+ T cells completely reserved the tumor-suppressive effect of WSX1, indicating that reinvigoration of CD8+ T cell activity was the indispensable mechanism underlying WSX1-induced tumor regression'. However, 'in vivo depletion of CD8+ T cells' does not actually modulate the 'reinvigoration of CD8+ T cell activity'.

14. Fig. 4a-b: It is confusing to note that the description of figure legend (analysis in tumor cells) is not consistent with that in the main text (a high expression of PD-L1 and p-AKT in hepatocyte). Did the changes in PD-L1, WSX1 and p-AKT expression occur in tumor cells or hepatocytes? Special tumor markers should be used to indicate the tumor cells and hepatocytes clusters, as their expression patterns are very different. For examples, the majority of the cells in Fig. 4a did not express WSX1. Were the WSX1-expressing cells tumor cells? Or hepatocytes? Is the negative relationship between WSX1 and p-AKT/PD-L1 in Fig. 4b represent the overall pattern or cluster-dependent?

15. What is the expression pattern of Ki-67 in these tumor cell/hepatocyte clusters? If the tumor suppressive effect of WSX1 is non-tumor cell-autonomous as claimed by the authors, there should be no difference in Ki-67 expression in the parenchymal cells between the 'Oncogene + WSX1' and 'Oncogene' groups.

16. Fig. 4d: What is the rationale of using SNU449 and SNU475 for in vitro assay? What are the WSX1 protein levels in these two cell lines compared to normal liver?

17. Fig. 4e: WSX1 downregulates PD-L1 expression on tumor cells, why was PD-L1+ tumor cell proportion changed? Same question for Fig. 5d, 5g, 6d, 6f.

18. Fig. 4g: It is inaccurate to use CD3 status as marker of tumor cell survival. This data also cannot explain that the 'sensitized T cell killing' effect is mediated by PD-L1 expression of tumor cells.

19. Fig. 5: The molecular characterization of the effect of WSX1 on PD-L1 degradation is comprehensive, but was solely characterized by ectopic expression in HCC cell lines, which may not be pathophysiological. The authors should utilize immortalized liver cells with WSX1 expression, and characterize its molecular action on PD-L1 via RNA interference-mediated knockdown or CRISPR/Cas9-mediated deletion.

20. Fig. 6: How does WSX1 reduce AKT phosphorylation? Is it possible that WSX1 regulates the expression level of chemokines and attract CD8+ T cell infiltration?

21. If WSX1 enhanced GSK β activity through inactivating AKT signaling, why don't the perturbation of this general oncogenic pathway affect the tumor-intrinsic phenotypes? It has been widely reported that AKT inhibition would reduce HCC cell proliferation, survival and invasion abilities.

22. Are there any correlation between WSX1 and PD-L1 in HCC cells, as well as PD1+CD8+T cells in HCC patients?

Minor comments:

1. The cluster labelling in Fig. 3a is not clear.
2. The KEGG pathway items in Fig. 6b are too small.
3. Discussion: "However, WSX1 expression in livers induced an increased infiltration of tumor-infiltrating lymphocytes and reduced CD8+ T cell dysfunction by downregulating PD-1, LAG-3, and CTLA-4." The effect of WSX1 on LAG-3 and CTLA-4 has not been investigated in this study.
4. Discussion: "In our study, WSX1 injection resulted in a remarkable upregulation of WSX1 on tumor cells without impacting its expression in infiltrating CD8+ T cells, showing that WSX1-rescued CD8+ T cell immunity is more likely due to its direct modification on tumor cells." The expression of WSX1 in infiltrating CD8+ T cells upon WSX1 injection has not been shown.

Reviewer #2 (Remarks to the Author): expertise in mechanisms of PD-L1 expression regulation.

The authors seek to understand the molecular mechanism underlying how WSX1 functions as a tumor suppressor to destabilize PD-L1 via reducing the AKT signaling to prevent CD8+ lymphocytes exhaustion. The paper is clearly written, however, the following concerns should be addressed before its publication at Nature Communications.

1. Figure 1, the authors should comment or explain how WSX1 expression is reduced in HCC, due to genetic deletion or due mRNA reduction?
2. Figure 2A, given that WSX1 can function as a IL27 receptor, it will be nice to side by side compare whether IL27-binding deficient WSX1 can suppress tumorigenesis as WT-WSX1.
3. Figure 4C, it will be important to include pAKT and downstream substrates in. It is also critical to show whether depletion of endogenous WSX1, on the other hand, can elevate pAKT and PD-L1.
4. Figure 4F, it will be important to use CHX chase to examine if WSX1 affects PD-L1 half-life.
5. Figure 5B, the labeling is off, hard to follow.
6. Figure 5C, will depletion of endogenous WSX1 reduce PD-L1 ubiquitination in cells, to stabilize PD-L1, whereas overexpression of WSX1 should destabilize PD-L1 by enhancing its ubiquitination in cells?
7. Figure 5F, it is important to monitor pS473AKT, as well as the phosphorylation status of known AKT substrates such as TSC2 or FOXO.
8. Figure 5H-J, it will be important to examine whether AKT inhibitor can reduce PD-L1 half-life and protein abundance while GSK3 inhibitor can stabilize PD-L1 by impairing PD-L1 ubiquitination to extend its half-life.
9. Figure 6C, as shown in Figure 6A-B, 80% of the PI3K/Akt pathway expression was found to changed upon WSX1 expression. However, in all the protein tested in Figure 6C, there is no major changes after overexpressing WSX1. The authors should validate their conclusions in Figure 6A-B by looking at particularly the genes being affected.
10. Figure 6I: In addition, it will be critical for the authors to identify the exact molecular mechanism of why and how WSX1 can reduce Akt signaling.

Reviewer #3 (Remarks to the Author): expertise in T cell signalling; GSK3b

The paper outlines the role of XSX1 in the control of PDL1 expression and CD8 responses against tumors. It is an interesting, has important results and is generally well performed paper. Several major issues should be addressed in the paper.

Specific Points

- 1) Fig. 2: The injection of WSX1 is impressive but man controls are missing. 1) an irrelevant control; 2) evidence of WSX1 expression is missing. 1) is needed to control for non-specific effects of infection similar to oncolytic virus'.
- 2) Does WSX1 directly affect the AKT-Ras pathway in HCC development?
- 3) Fig. 3: viSNE plots look like reduced T- and NK cells? The figures desparately need increased

size text since it is too hard to read. Where is the CD4 and CD8 staining in viSNE plots...this is needed to help reader to follow the data. This is unnecessarily hard to follow and needs to be made reader friendly. (i) Reduced PD-1, LAG3 expression but oddly also reduced activation markers Ki67 and GZMB with oncogene plus WSX1...why is this? Are effector T-cells altered and less effective?

4) The CyTOF data is poorly presented. Fig. 3b: what is the marker chosen (anti-CD3?). However again it does not look correct based on the viewing of panel a. The author needs to show absolute numbers, if possible in responders vs non-responders.

Panel d is poorly described...the text needs to name the subsets in terms of their surface markers etc. and what this means potentially in terms of expected functional outcomes. Fig. 3c...again needs to describe the phenotype of the cluster and make sense as to how this may be targeted in terms of function. Panel h: did they look at TOX expression the most reliable marker for exhaustion.

5) Figure 4: nice findings.

6) Figure 6: is somewhat problematic since it is also probably needing other mediators calcium etc. Where is the surprise and what makes the claim specific? How does GSK-3 fit into this based on the data? Other signaling proteins need to be examined. DN AKT needs to be examined to prove that the pathway is essential. Reconstitution analysis would be helpful to show specificity by co-transfection. I would delete figure since it is an overinterpretation.

Point-to-point Response to Reviewers' comments:

Reviewer #1 (Remarks to the Author): expertise in HCC

In this study, the authors investigated the mechanism by which tumor cells escape from host immunosurveillance and examined the role of a potential tumor suppressor gene, WSX1, on HCC development. Mechanistically, they showed that WSX1 induces PD-L1 degradation via AKT/GSK3 β pathway in tumor cells, which may in turn reduce CD8+T cell functional exhaustion via PD-1/PD-L1 interaction to suppress tumor development. While these findings are of scientific values, some of the findings are not well supported by the current data.

Major comments:

1. Fig. 1a: The expression of WSX1 was examined in a human multiple normal tissue microarray. Most of the 33 types of normal tissues showed high level of staining signal, including the liver whose signal was even higher than in immune cell-enriched tissues. How is the specificity of the antibody against WSX1 verified? The authors must provide negative controls for the immunostaining e.g. using antigen-absorbed antibody, otherwise the authenticity of the expression pattern in normal and malignant tissues is in doubt.

Thanks for your kind advice. The antibody we used in the human tissue microarray is a commercial antibody against WSX1 (ThermoFisher Scientific, PA5-19984). To verify the specificity of the antibody, we have performed 2 experiments as follows:

- (1) We used a human WSX1 synthetic antigen (ThermoFisher Scientific, PEP-0108) as an antigen absorbed control, which totally blocked the tissue WSX1 antigen binding of WSX1 antibodies.
- (2) We confirmed the IHC staining results by using another WSX1 monoclonal antibody (clone 237) produced by the Monoclonal Antibodies Core Facility at MD Anderson Cancer Center, which we detailed in our previous study.¹

We hope that these results will be sufficient for verifying the specificity of our staining results.

2. Fig. 1c: According to the authors, there were 130 cases of HCC in the human liver tissue microarrays (BC03116a and HLiv-HCC180Sur-03). But why “HCC patients were manually divided into low (n = 47) and high (n = 43) WSX1 expression groups”? How about the other 40 cases that were excluded from the survival analysis? Moreover, what is the expression level of WSX1 and its relationship with patient survival in TCGA data?

Among those 130 cases of human HCC, only 90 cases (HLiv-HCC180Sur-03) had the desired survival information for this analysis; the other 40 cases lacked survival information, forcing us to exclude them from the survival analysis. We have clarified this in both the Methods and Results sections of our revised manuscript.

Our analysis of the role of WSX1 was based on protein expression, not mRNA, levels. However, we did perform a survival analysis based on mRNA levels, as the reviewer suggested, but we did not find a significant association of WSX1 mRNA level with overall survival ($P = 0.2676$).

3. Fig. 2b, 2f, 3k, s2b, s2e, s4b: Tumor weight, tumor volume or number of tumor nodule instead of liver weight should be shown for accurate assessment of tumorigenicity.

Unlike an orthotopic implantation tumor model, the tumor models we used in this study are novel spontaneous HCC mouse models. Instead of inoculating tumor cells, we delivered plasmid DNA encoding *NRAS* and *AKT* into mouse livers via hydrodynamic injection into the tail vein to transform hepatocytes. As described previously, after hydrodynamic transfection, a large number of hepatocytes are transfected, and all these cells can potentially produce tumors.^{2,3} This large number of cell transfections yields numerous large tumor nodules throughout the mouse liver—too many to be counted.^{2,3} Moreover, in this model, there are no visible explicit boundaries at which to distinguish tumor lesions, preneoplastic areas, and normal liver tissues. Therefore, measurements of tumor weight, tumor volume, or number of tumor nodules might not be ideal for our study, though we fully understand and respect this reviewer's suggestion.

To address this reviewer's question and make the assessment of tumorigenicity more accurate, we have added statistical analyses of quantitation of H&E staining—percentages of area containing preneoplastic/tumor lesions (Figure 2d and 2i), in which hepatocytes have cytoplasmic basophilia and a high nuclear-to-cytoplasmic ratio.

4. Fig. 3a and 4a: The tissues used for MS-CyTOF analysis were not clearly described. It was mentioned in the main text that 'intrahepatic infiltrating immune cells' were isolated for Fig. 3. However, in the figure legend for Fig. 4, 'tumor cells isolated from mice livers' were analyzed. As the immune cell profiles and gene expressions between tumor and adjacent liver tissues would be vastly different, the authors should examine the immune cell profiles and gene expressions in both tumor and liver tissues separately and explicitly mention the tissue identity involved in the data.

Thanks for your valuable advice. We have rewritten the result for Fig. 3 to ensure clarity. To further clarify, the MS-CyTOF analysis shown in Fig. 3a is of intrahepatic infiltrating immune cells, while the analysis in Fig. 4a is of hepatic parenchymal cells (including hepatocytes/tumor cells in different transformation stages). They were both obtained from whole mouse livers in the mouse model described in Fig. 2a. Specifically, single-cell suspensions were prepared from whole mouse livers and then divided into immune cells and hepatocytes/tumor cells by density gradient centrifugation and a FACSaria Cell

Sorter (BD Biosciences). We have explicitly stated the tissue identity in the Results section of our revised manuscript.

As mentioned in the response to question 3, due to the characteristics of the spontaneous HCC model, there is no visible explicit boundary between tumor and liver tissues. Therefore, we were unable to isolate immune cells in tumor and adjacent liver tissues separately for separate immune profiling.

5. Fig. 3a: The authors stated that 'the majority is T cells. How to define 'majority', by cell number or percentage or cluster numbers? Since the size of the tumor and liver tissues in the 'Oncogene' and 'Oncogene + WSX1' groups are very different as shown in Fig. 2a, are the differences observed in Fig. 3a due to WSX1 expression or tissue difference? Without clear description of tissues involved, the meaning of the data is not clear.

Our statement that the majority is T cells was based on the proportion of T cells among total intrahepatic immune cells (CD45-positive cells).

All tissues shown in Figure 2a are the entire liver obtained from each mouse. The difference in liver size was due to WSX1 expression. Likewise, the differences observed in Fig. 3a were due to WSX1 expression, since the intrahepatic immune cells we analyzed were isolated from the same type of tissues—the whole liver of each mouse. To make this clearer, we have added better descriptions of each type of tissue in our revised manuscript.

6. Fig. 3b and 3c: Notably, WSX1 treatment significantly increased the proportion of infiltrating T cells from 41.43% ± 5.87% to 54.27% ± 1.76% (P = 0.0221, n = 3, Figure 3b), while WSX1 had no significant effect on the proportion of CD4+, CD8+, and CD4+CD8+ T cells (Figure 3c). This statement is confusing. Where were T cells infiltrating to? Tumor or liver tissues?

The proportion of infiltrating T cells in Figure 3b represented the percentage of CD3+ T cells among total intrahepatic CD45+ immune cells, while the proportions of CD4+, CD8+, and CD4+CD8+ T cells in Figure 3c represented the percentages among all intrahepatic CD3+ T cells. To make this clear, we have clarified our statements in the revised manuscript.

The T cells we analyzed were T cells infiltrating the whole liver tissues—intrahepatic T cells. As we stated above, due to the unique characteristics of the spontaneous HCC model, there is no visible explicit boundary between tumor and normal liver tissues. In addition, the whole hepatic immune microenvironment was important for HCC initiation and progression, so we analyzed intrahepatic T cells as a whole.

7. Why is the proportion of T cells increased if the key function of WSX1 is to maintain T cell activity through down-regulating PD-L1?

Thanks to your kind reminder, we reanalyzed our data regarding the proportion of T cells. The proportion of entire T cells was impacted by the number of other T cell types such as NKT and $\gamma\delta$ T cells. When all T cells were included, we found that WSX1 had no significant effect on the proportion of T cells among the entire population of immune cells (CD45-positive cells). We have clarified our statement in the revised manuscript.

8. Fig. 3d: Which tissue does the heatmap of T cell panel markers correspond to, 'Oncogene' or 'Oncogene + WSX1'? The heatmaps of both groups should be shown.

Entire livers embedded with diffuse tumors were used for this analysis. The MS-CyTOF analysis identified similar cell clusters based on the expression patterns of cell markers in all samples, with the goal of detecting differences in cell-subset abundance between groups. Therefore, the heatmaps for cell marker expression are much the same in both groups, which are shown in the revised Figure 3.

9. For the comprehensive characterization of T cell subtypes, gamma-delta T and NKT cells should also be included in the antibodies panel for CyTOF.

As stated in Supplementary Table 1, we included TCR $\gamma\delta$, CD3, CD25, and CD69 in the antibody panel for $\gamma\delta$ T cells and NKp46, NK1.1, CD3, and CD44 for NKT cells. Based on our MS-CyTOF results, we found that WSX1 treatment had no significant effect on the frequencies of either $\gamma\delta$ T cells or NKT cells.

10. Fig. 3h: The authors used PD1+LAG3+CTLA4+ to define exhausted CD8+T cells, and PD1-LAG3-CTLA4- to define non-exhausted CD8+T cells. However, as one of the important markers of functional exhausted CD8+T cells, Tim3 was not included in the analysis.

Thanks for your good question. Tim3 was omitted because only a few Tim3-positive cells, as shown below, were detected in our model, as determined by flow cytometry analysis. Therefore, we did not include Tim3 in our antibody panel for MS-CyTOF analysis.

11. Fig. 3i: How to explain the reduction of GranB and Ki67 in the 'Oncogene + WSX1' group when compared to the 'Oncogene' group? These activated T cell markers are not supposed to reduce together with other inhibitory markers. This data casts doubt on whether WSX1 reduce T cell exhaustion or T cell function?

Thanks for your advice and sorry for the confusion. The expression intensity of GranB and Ki67 listed in the original Fig. 3I is the total expression level for all T cells, which was obviously not suitable for interpreting the biological function of the critical population of T cells—the CD8+ T cell subset, which we found to be crucial for HCC progression in our study. Therefore, we performed the suggested analysis against the CD8+ subset of interest independently via multicolor flow cytometry and found a significant increase in both GranB and Ki67 ($p < 0.05$). These results are shown in Supplementary Figure S5 in the revised manuscript.

12. Fig. 3: More importantly, the CyTOF data needs to be independently validated by multi-color flow cytometry using specific subsets of exhaustion and functional markers. This will greatly help to resolve the confusing points as listed above.

Done. We independently validated our CyTOF results by analyzing exhaustion (PD-1/LAG-3/CTLA-4) and functional markers (Granzyme B/Perforin/Ki67) using multicolor flow cytometry. As shown in the revised Supplementary Figure S5, WSX1 induced reduction of exhaustion markers (PD-1/LAG-3/CTLA-4) and increase of functional markers (Granzyme B/Perforin/Ki67) in CD8+ T cells.

13. Fig. 3j-L: The data of the various immune cell depletion was impressive. Yet, the levels of immune cells (CD8T, CD4T and NK) should be determined to confirm the depletion. The authors concluded that ‘Intriguingly, either adopting immune-deficient mice or in vivo depletion of CD8+ T cells completely reserved the tumor-suppressive effect of WSX1, indicating that reinvigoration of CD8+ T cell activity was the indispensable mechanism underlying WSX1-induced tumor regression’. However, ‘in vivo depletion of CD8+ T cells’ does not actually modulate the ‘reinvigoration of CD8+ T cell activity’.

Completed. We have measured the levels of immune cells (CD8+ T, CD4+ T, and NK cells) to confirm the efficiency of immune cell depletion. All results are included in the revised Figure 4a.

As you said, the results of in vivo immune cell depletion could not actually prove the “reinvigoration of CD8+ T cell activity,” but it might be sufficient to support our hypothesis that the tumor-suppressive effect of WSX1 predominantly relied on the existence of CD8+ T cells. Therefore, we have corrected our statements in our revised manuscript.

14. Fig. 4a-b: It is confusing to note that the description of figure legend (analysis in tumor cells) is not consistent with that in the main text (a high expression of PD-L1 and p-AKT in hepatocyte). Did the changes in PD-L1, WSX1 and p-AKT expression occur in tumor cells or hepatocytes? Special tumor markers should be used to indicate the tumor cells and hepatocytes clusters, as their expression patterns are very different. For examples, the majority of the cells in Fig. 4a did not express WSX1. Were the WSX1-expressing cells tumor cells? Or hepatocytes? Is the negative relationship between WSX1 and p-AKT/PD-L1 in Fig. 4b represent the overall pattern or cluster-dependent?

We apologize for the confusion. In the revised manuscript, we have consistently named all those cells as hepatocytes, including both nontransformed and transformed hepatocytes in different transformation stages. Unfortunately, we were unable to distinguish transformed from nontransformed hepatocytes for the following reasons: (1) Unlike traditional orthotopic implantation tumor models using direct tumor cell implantation, our tumor model is a spontaneous HCC mouse model, in which tumors were induced by hydrodynamic injection of plasmids encoding *NRAS* and *AKT*. As previously reported, the injected plasmid DNA is atypically distributed into more than 90% of hepatocytes, at least 1% to 2% of

hepatocytes are transfected, and all these cells can potentially yield tumors.^{2,3} (2) There are no visible explicit boundaries among tumor cells, preneoplastic cells, and normal hepatocytes due to the widespread tumor cells, as shown in Figure 2. (3) Although past efforts have led to the identification of multiple potential tumor biomarkers for HCC, there is no single recurrent biomarker that can identify all HCCs. No specific tumor marker could clearly distinguish normal hepatocytes and heterogeneous HCC cells in our spontaneous mouse models. In our study, most of the nontransformed hepatocytes were WSX1-positive cells, while tumor cells were either WSX1 negative or low. However, WSX1 expression is not a specific HCC biomarker because we found WSX1 expression was also downregulated in severely cirrhotic liver tissues, as shown below.

Regardless, the negative relationship between WSX1 and p-AKT/PD-L1 (revised Figure 7d) represents the overall pattern in all hepatocytes, including transformed and nontransformed hepatocytes.

15. What is the expression pattern of Ki-67 in these tumor cell/hepatocyte clusters? If the tumor suppressive effect of WSX1 is non-tumor cell-autonomous as claimed by the authors, there should be no difference in Ki-67 expression in the parenchymal cells between the ‘Oncogene + WSX1’ and ‘Oncogene’ groups.

As shown below, no significant difference of Ki-67 expression was observed in hepatocytes between the “Oncogene + WSX1” and “Oncogene” groups ($P=0.8788$).

16. Fig. 4d: What is the rationale of using SNU449 and SNU475 for in vitro assay? What are the WSX1 protein levels in these two cell lines compared to normal liver?

As stated in Supplementary Figure S3a, SNU449 and SNU475 cells had the lowest expression level of WSX1 among 5 human HCC cell lines (SNU398, SNU449, SNU475, Hep3B, and HepG2), so we chose these two HCC cell lines to check the impact of ectopic expression of WSX1.

To compare the WSX1 protein levels in SNU449 and SNU475 to normal liver, we adopted THLE-2 cells, which were derived from primary normal liver cells by infection with SV40 large T antigen, as a relative normal control. Our results showed that the WSX1 protein levels in these two cell lines were significantly lower than in THLE-2 (Supplementary Figure S3a). Those results were consistent with our findings in human liver tissues, in which WSX1 was highly expressed in normal liver tissues but decreased to different degrees in HCC tissues.

17. Fig. 4e: WSX1 downregulates PD-L1 expression on tumor cells, why was PD-L1⁺ tumor cell proportion changed? Same question for Fig. 5d, 5g, 6d, 6f.

The percentage of PD-L1⁺ HCC cells, which was analyzed using flow cytometry, represents the cell surface expression level of PD-L1 on HCC cells. Western blotting results represent the total protein level of PD-L1. Our study found that WSX1 reduced both cell surface expression and total protein levels of PD-L1. The decrease in the PD-L1⁺ proportion of HCC cells, which represents downregulation of cell surface PD-L1, might be a result of the reduction in total PD-L1 protein production. Another explanation is that the quantity of the injected WSX1 plasmid DNA may not have been evenly distributed across all hepatocyte cells, resulting in variation in expression of WSX1 and the associated variation of surface PD-L1 expression among HCC cells.

18. Fig. 4g: It is inaccurate to use CD3 status as marker of tumor cell survival. This data also cannot explain that the 'sensitized T cell killing' effect is mediated by PD-L1 expression of tumor cells.

In fact, CD3 is not used as a biomarker for tumor cell survival in this study (sorry for any lack of clarity). In Fig. 4g, CD3, which is a well-known T-cell biomarker, was used to distinguish human T cells from HCC cells. In the T cell-mediated tumor cell killing assay, after coculture, we used a SYTOX™ AADvanced™ Dead Cell Stain Kit to exclude dead cells and then used anti-human CD3a-specific antibodies to distinguish live T cells from live HCC cells. Since the ratio of tumor cells to T cells is fixed in both groups at the beginning, a decreased ratio of HCC to T cells represents the tumor cell-killing activity of human T cells. Several previous reports used this assay to detect T-cell activity.⁴

We agree that these data alone cannot demonstrate that the 'sensitized T cell killing' effect is mediated by PD-L1 expression of tumor cells. We have clarified our statements in the Results section in the revised manuscript.

19. Fig. 5: The molecular characterization of the effect of WSX1 on PD-L1 degradation is comprehensive but was solely characterized by ectopic expression in HCC cell lines, which may not be pathophysiological. The authors should utilize immortalized liver cells with WSX1 expression and characterize its molecular action on PD-L1 via RNA interference-mediated knockdown or CRISPR/Cas9-mediated deletion.

An *in vivo* model was included in this revision. To respond to this question, we established WSX1-deficient (WSX1^{-/-}) mice, in which the physiological level of WSX1 was eliminated. As shown below, we examined PD-L1 expression in liver tissue lysates from both wild-type and WSX1^{-/-} mice and found that most of the WSX1^{-/-} mice had higher levels of PD-L1 than the wild-type mice. Moreover, we knocked down WSX1 in SNU398 cells, which have a relatively high expression level of WSX1. We found that WSX1 knockdown reduced PD-L1 expression in SNU398 cells (Figure 5f and 5g, and Supplementary Figure S7).

20. Fig. 6: How does WSX1 reduce AKT phosphorylation? Is it possible that WSX1 regulates the expression level of chemokines and attract CD8⁺ T cell infiltration?

A novel mechanism was identified. WSX1 is a novel protein, and no literature is available regarding its effects on nonimmune cells or the underlying signaling pathways mediating its intrinsic effects. In the present study, we discovered that WSX1 reduced PD-L1 expression by promoting GSK3 β activity through inhibiting AKT phosphorylation without affecting total AKT levels (Figure 7a). To investigate how WSX1 reduces AKT phosphorylation, we focused our effort on its regulator. As we know, multiple mechanisms control AKT phosphorylation, among which PI3K and PTEN are known as the most critical regulators. Thus, we examined the effects of WSX1 on PI3K and PTEN protein levels and found that WSX1 significantly reduced PI3K-p100 δ expression, while no significant difference was found in levels of PTEN, PI3K-p100 α , or PI3K-p85 (revised Figure 7e). In support of this novel observation, our data showed that WSX1 overexpression reduced PI3K-p100 δ (*PIK3CD*) mRNA levels, which might be responsible for the downregulation of PI3K-p100 δ protein levels. In other words, WSX1-mediated inhibition of AKT phosphorylation is not regulated by the well-known classical PI3K isoforms PI3K α or PI3K β , which are ubiquitously expressed, but by the PI3K δ isoform, which was reported to play significant roles in malignant liver tumors and correlated with poor survival rates for HCC patients.⁵ We have included this novel revelation in the revised manuscript. We are excited about this novel discovery, but the detailed mechanism will require another one or two manuscripts to uncover.

As mentioned above, the intrinsic effects of WSX1 are almost an unexplored area. Based on our current discoveries, WSX1 reduces PI3K δ transcription to inhibit AKT. We do not exclude other possible mechanisms such as the one suggested by this thoughtful reviewer, but we do not have any data to support them yet.

21. *If WSX1 enhanced GSK β activity through inactivating AKT signaling, why don't the perturbation of this general oncogenic pathway affect the tumor-intrinsic phenotypes? It has been widely reported that AKT inhibition would reduce HCC cell proliferation, survival and invasion abilities.*

Agreed, direct perturbation of the AKT oncogenic pathway would affect HCC cell proliferation, survival, and invasion abilities. However, the perturbation in the AKT oncogenic pathway induced by WSX1 is different from the classical one. In our case, WSX1 inhibits AKT pathway through reducing PI3K δ without impacting other isoforms of PI3K, which is often observed in the published data. This unique

perturbation of AKT in transforming or transformed HCC cells impacts PD-L1 expression and immune surveillance for controlling tumor initiation/progression—a novel observation revealed for the first time in this MS. These novel observations will add novel thinking about the design of in vivo and clinical experiments using AKT perturbation for HCC prevention or treatment.

22. *Are there any correlation between WSX1 and PD-L1 in HCC cells, as well as PD1+CD8+T cells in HCC patients?*

Yes, the negative correlation between WSX1 and cell surface PD-L1 is definitive, as shown in our MS through multiple lines of evidence. We are not sure about the second part of the question from this reviewer because our central discovery is about WSX1’s negative regulation of PD-L1 expression in hepatocytes or HCC cells to inhibit exhaustion of CD8+ T cells, which show high expression of inhibitory receptors such as PD-1. It is extremely challenging to simultaneously quantitate WSX1 and PD-L1 expression in tumor cells, as well as cell surface PD-1+CD8+T cells in patient HCC tissues. However, there is a trend of increased expression of total PD-L1 and PD-1 in WSX1-low cancers (36.7%) vs WSX-high cancers (10.0%) in a 30-patient IHC analysis. However, these results cannot be overinterpreted. Fresh HCC samples will serve this purpose best, but we do not have access to them yet.

	WSX1 ^{high}	WSX1 ^{low}
PD-L1 ^{high} PD-1 ^{high}	10.0%	36.7%
PD-L1 ^{low} PD-1 ^{low}	20.0%	16.7%
PD-L1 ^{high} PD-1 ^{low} /PD-L1 ^{low} PD-1 ^{high}	3.3%	13.3%
Total	33.3%	66.7%

Minor comments:

1. *The cluster labelling in Fig. 3a is not clear.*

Done. Thanks for your kind reminder. We have clarified it in the revised manuscript.

2. *The KEGG pathway items in Fig. 6b are too small.*

Done. Thanks for your kind reminder. We have made the suggested modification in the revised manuscript.

3. *Discussion: “However, WSX1 expression in livers induced an increased infiltration of tumor-infiltrating lymphocytes and reduced CD8+ T cell dysfunction by downregulating PD-1, LAG-3, and CTLA-4.” The effect of WSX1 on LAG-3 and CTLA-4 has not been investigated in this study.*

Done. Thanks for your kind reminder. We have corrected the statement to “However, WSX1 expression in livers reduced CD8+ T cell dysfunction mainly by downregulating PD-1” in the revised Discussion. We intend to explore the effect of WSX1 on LAG-3 and CTLA-4 in future work.

4. *Discussion: “In our study, WSX1 injection resulted in a remarkable upregulation of WSX1 on tumor cells without impacting its expression in infiltrating CD8+ T cells, showing that WSX1-rescued CD8+ T cell immunity is more likely due to its direct modification on tumor cells.” The expression of WSX1 in infiltrating CD8+ T cells upon WSX1 injection has not been shown.*

Done. Thanks for your kind reminder. We have added the results in revised Supplementary Figure 5a.

Reviewer #2 (Remarks to the Author): expertise in mechanisms of PD-L1 expression regulation.

*The authors seek to understand the molecular mechanism underlying how *WSX1* functions as a tumor suppressor to destabilize PD-L1 via reducing the AKT signaling to prevent CD8+ lymphocytes exhaustion. The paper is clearly written, however, the following concerns should be addressed before its publication at Nature Communications.*

*1. Figure 1, the authors should comment or explain how *WSX1* expression is reduced in HCC, due to genetic deletion or due mRNA reduction?*

To reveal how *WSX1* expression is reduced in HCC, we first analyzed TCGA data. As shown in the following figure, although *WSX1* protein expression levels were significantly decreased in HCC patients, no difference in *WSX1* mRNA was observed between normal livers and HCCs. In addition, according to the data on COSMIC (Catalogue of Somatic Mutations in Cancer, https://cancer.sanger.ac.uk/cosmic/gene/analysis?all_data=&coords=AA%3AAA&dr=&end=637&gd=&id=230594&ln=IL27RA&seqLen=637&sn=liver&start=1#ts), among 901 HCC cases, only 7 (0.78%) had point mutations in *WSX1*.

Based on the above observation, the reduction of *WSX1* in HCC is not at the transcriptional level but is controlled at the protein level. Posttranslational modification and protein stability may serve as critical mechanisms, and this will require a great deal of work to figure out in the future. Our current MS focused on how *WSX1* regulates cell surface PD-L1 expression and the associated change in immune surveillance to impact HCC initiation/progression.

2. *Figure 2A, given that WSX1 can function as an IL27 receptor, it will be nice to side by side compare whether IL27-binding deficient WSX1 can suppress tumorigenesis as WT-WSX1.*

Done. The IL27 binding site for WSX1 is not crystal clear, and mutation or depletion of the candidate binding fragment could damage the biological function of WSX1. To address this excellent question without damaging the full biological function of WSX1, the easiest way is to use the ligand-deficient mouse model, in which no ligand of WSX1 is present to impact its activity. Therefore, we used IL-27p28^{-/-} C57BL/6J mice, since p28 is known to be the indispensable binding subunit of IL27 for WSX1. As shown in our results, in the context of deficient IL27 binding, WSX1 retains its full biological function in impairing *NRAS/AKT* oncogene-induced HCC formation and improving overall survival in IL-27p28^{-/-} mice (Supplementary Figure S2).

3. *Figure 4C, it will be important to include pAKT and downstream substrates in. It is also critical to show whether depletion of endogenous WSX1, on the other hand, can elevate pAKT and PD-L1.*

Agreed. Indeed, pAKT and its downstream substrates are included in revised Fig. 6g and Fig. 7a.

For the second question, we knocked down WSX1 in SNU398 cells. We found that WSX1 knockdown reduced both pAKT and PD-L1 expression in SNU398 cells (Figure 5 and Supplementary Figure S7).

4. *Figure 4F, it will be important to use CHX chase to examine if WSX1 affects PD-L1 half-life.*

Done. As suggested, we analyzed the effects of WSX1 on PD-L1's half-life and included the results in the revised Figure 6f and Supplementary Figure S7c.

5. *Figure 5B, the labeling is off, hard to follow.*

Done. Thanks for your kind reminder. We have corrected it in the revised figure.

6. *Figure 5C, will depletion of endogenous WSX1 reduce PD-L1 ubiquitination in cells, to stabilize PD-L1, whereas overexpression of WSX1 should destabilize PD-L1 by enhancing its ubiquitination in cells?*

Done. As mentioned before, we performed RNA interference-mediated reduction of endogenous WSX1 in SNU398 cells and determined its effect on PD-L1 ubiquitination. The results are shown in the revised manuscript (Supplementary Figure S7).

7. *Figure 5F, it is important to monitor pS473AKT, as well as the phosphorylation status of known AKT substrates such as TSC2 of FOXO.*

Done. There are well over 100 AKT substrates reported in the research literature, among which GSK3 β was the first to be reported. pS473AKT exerts inhibitory phosphorylation on an amino-terminal motif conserved in GSK3 β (Ser9). As you suggested, besides pGSK3 β ^{Ser9}, we analyzed the phosphorylation status of TSC2 as well. Reduction of p-TSC2 further supported the inhibition of AKT activity (Figure 7a).

8. *Figure 5H-J, it will be important to examine whether AKT inhibitor can reduce PD-L1 protein abundance while GSK3 inhibitor can stabilize PD-L1 by impairing PD-L1 ubiquitination to extend its half-life.*

Several studies have reported effects of AKT or GSK3 inhibitors on PD-L1 protein expression.⁶ A previous report claimed that knockdown of GSK3 β increased PD-L1 expression.⁴ Inhibition of GSK3 β by

LiCl prevented osimertinib-induced PD-L1 degradation.⁷ AKT inhibitors were also reported to downregulate PD-L1 in tumor cells.^{8,9} However, no literature has reported the effect of AKT inhibitors on PD-L1 ubiquitination. In our study, we found that the GSK3 inhibitor LiCl indeed upregulated PD-L1 (Figure 6h and 6i) and reduced PD-L1 ubiquitination in HCC cells (shown in the following figure, panel a). Although we found that the AKT inhibitor MK-2206 slightly reduced PD-L1 expression (shown in the following figure, panel b), it cannot mimic WSX1-induced PD-L1 destabilization. This difference is perhaps due to the functional difference between WSX1-mediated inhibition of AKT and the chemical inhibitor-mediated inhibition of AKT. AKT inhibitors have been reported to repress PD-L1 transcription,⁸ but WSX1 had no obvious influence on PD-L1 mRNA. As shown in our MS, WSX1 induced AKT inhibition through reducing PI3K δ without impacting other classical isoforms of PI3Ks, indicating a unique regulatory characteristic of WSX1 on AKT activity. The chemical inhibitors result in a direct perturbation of the AKT oncogenic pathway via affecting the tumor-intrinsic phenotypes.¹⁰ However, WSX1-induced AKT inhibition had no significant effect on HCC proliferation and migration. Isoform-specific effects of the individual PI3Ks on AKT activity might be one of the mechanisms responsible for this difference, but the underlying mechanism needs further exploration. These functional differences suggest that these AKT chemical inhibitors are not ideal for use in validating WSX1-mediated PD-L1 reduction.

9. Figure 6C, as shown in Figure 6A-B, 80% of the PI3K/Akt pathway expression was found to changed upon WSX1 expression. However, in all the protein tested in Figure 6C, there is no major changes after overexpressing WSX1. The authors should validate their conclusions in Figure 6A-B by looking at particularly the genes being affected.

We apologize for the misunderstanding caused by our unclear descriptions. We did not mean that 80% of genes in the entire PI3K/AKT pathway were changed upon WSX1 expression. According to the KEGG pathway enrichment analysis of our RNA-Seq data, about 30 differentially expressed genes induced by WSX1 were enriched in the PI3K/AKT pathway, and 80% of those 30 genes were downregulated. Among those downregulated genes, *PIK3CD*, encoding the PI3K-p100 δ protein, was reported to play a critical role in both AKT activation and HCC progression. Our results show that WSX1 reduces both protein and mRNA levels of PI3K-p100 δ in HCC cells (Figure 7e and 7g). In conclusion, WSX1 transcriptionally downregulates PI3K-p100 δ , thereby reducing AKT activation and subsequently promoting GSK3 β -mediated PD-L1 degradation.

10. Figure 6I: In addition, it will be critical for the authors to identify the exact molecular mechanism of why and how WSX1 can reduce Akt signaling.

As detailed in addressing question 20 for Reviewer 1, we have performed multiple additional experiments to address this question (see also Figure 7). In brief, WSX1 transcriptionally reduced PI3K δ (*PIK3CD*), which is a nonclassical isoform of PI3K that activates AKT and promotes HCC progression. But the more detailed mechanism will require another one or two manuscripts to uncover.

Reviewer #3 (Remarks to the Author): expertise in T cell signalling; GSK3b

The paper outlines the role of WSX1 in the control of PDL1 expression and CD8 responses against tumors. It is an interesting, has important results and is generally well performed paper. Several major issues should be addressed in the paper.

Specific Points

1) Fig. 2: The injection of WSX1 is impressive but main controls are missing. 1) an irrelevant control; 2) evidence of WSX1 expression is missing. 1) is needed to control for non-specific effects of infection similar to oncolytic virus'.

1) The crucial control vector was indeed included. The control vector without the gene of interest (WSX1) was used as a negative control for nonspecific effects of in vivo transfection, as previously reported.^{2,3} This control vector is also a plasmid DNA but not a virus, so there is no oncolytic virus effect.^{2,3} 2) WSX1 expression after DNA injection was measured; results are shown in Figure 2k.

2) Does WSX1 directly affect the AKT-Ras pathway in HCC development?

For the detailed response, please see the response to question 20 for Reviewer 1 and the revised Figure 7. In brief, our results showed that WSX1 reduced PI3K δ transcription, which subsequently downregulated AKT activity and reduced GSK3 β phosphorylation (Figure 7), but no significant effect of WSX1 on RAS was found, as shown in the figure below.

3) Fig. 3: viSNE plots look like reduced T- and NK cells? The figures desparately need increased size text since it is too hard to read. Where is the CD4 and CD8 staining in viSNE plots...this is needed to help reader to follow the data. This is unnecessarily hard to follow and needs to made reader freindly. (i) Reduced PD-1, LAG3 expression but oddly also reduced activation markers Ki67 and GZMB with oncogene plus WSX1...why is this? Are effector T-cell altered and less effective?

Done. Thanks for your valuable advice. We have increased the font size in the revised figures. No significant difference was observed in proportions of T- or NK cells (Figure 3b).

The expression intensity of GranB and Ki67 in the original Fig. 3I is the total expression level for all T cells, which is obviously not suitable for interpreting the biological function of the critical CD8+ T-cell subset in our study. Therefore, we have performed the suggested analysis against the CD8+ subset of interest and validated independently by multicolor flow cytometry. As shown in Supplementary Figure S5, WSX1 reduces the expression of exhaustion biomarkers (PD-1/LAG-3/CTLA-4) and increases the expression of immune surveillance functional markers (Granzyme B/Perforin/Ki67) in CD8+ T cells.

4) The CyTOF data is poorly presented. Fig. 3b: what is the marker chosen (anti-CD3?). However again it does not look correct based on the viewing of panel a. The author needs to show absolute numbers, if possible in responders vs non-responders.

Done. Thanks for your valuable advice. We have improved the figure presentation of CyTOF data. In Fig. 3b, anti-CD3 was chosen as a biomarker of T cells.

In Fig. 3a, the t-distributed stochastic neighbor embedding (t-SNE) map is derived from CyTOF analysis on equal numbers of intrahepatic immune cells in both groups, in which cells were colored by clusters identified by PhenoGraph and clusters were grouped by expression profiles of cell markers. In accordance with previous studies and the expertise of our bioinformatician, we used the proportions of cell subsets instead of absolute numbers to clarify immune cell profiles and perform the comparative analyses.

Panel d is poorly described...the text needs to name of subsets in terms of their surface markers etc. and what this means potentially in terms of expected functional outcomes. Fig. 3c....again needs to describe the phenotype of the cluster and make sense as to how this may be targeted in terms of function. Panel h: did they look at TOX expression the most reliable marker for exhaustion.

Done. As you suggested, it is better to show the name of each subset instead of the cluster number in Figures 3d, 3e, and 3f. For the CyTOF analysis, multiple immune cell subsets were identified based on expression of nearly 30 cell markers. However, not all immune cell clusters have elliptical names, thus, it might affect the text size in the figure and the readability if we were to present the full name of each cell cluster. Considering those issues mentioned above, expression levels of 28 T-cell panel markers in 13 T-cell clusters are shown in Figure 3e to address their identity.

As you said, TOX is also a good marker for exhaustion; however, until very recently, no antibody against TOX was available for CyTOF analysis, so we didn't determine TOX expression in this study.

5) Figure 4: nice findings.

Thanks for your comment.

6) Figure 6: is somewhat problematic since it is also probably needing other mediators calcium etc. Where is the surprise and what makes the claim specific? How does GSK-3 fit into this based on the data? Other signaling proteins need to be examined. DN AKT needs to be examined to prove that the pathway is essential. Reconstitution analysis would be helpful to show specificity by co-transfection. I would delete figure since it is an overinterpretation.

Done. Thanks for your suggestion; we have deleted the original Figure 6e and 6f.

References

1. Dibra, D., *et al.* IL27 controls skin tumorigenesis via accumulation of ETAR-positive CD11b cells in the pre-malignant skin. *Oncotarget* **7**, 77138-77151 (2016).
2. Ho, C., *et al.* AKT (v-akt murine thymoma viral oncogene homolog 1) and N-Ras (neuroblastoma ras viral oncogene homolog) coactivation in the mouse liver promotes rapid carcinogenesis by way of mTOR (mammalian target of rapamycin complex 1), FOXM1 (forkhead box M1)/SKP2, and c-Myc pathways. *Hepatology* **55**, 833-845 (2012).
3. Chen, X. & Calvisi, D.F. Hydrodynamic transfection for generation of novel mouse models for liver cancer research. *Am J Pathol* **184**, 912-923 (2014).
4. Li, C.W., *et al.* Glycosylation and stabilization of programmed death ligand-1 suppresses T-cell activity. *Nat Commun* **7**, 12632 (2016).
5. Ko, E., *et al.* PI3K δ Is a Therapeutic Target in Hepatocellular Carcinoma. *Hepatology* **68**, 2285-2300 (2018).
6. Wang, Y., *et al.* Regulation of PD-L1: Emerging Routes for Targeting Tumor Immune Evasion. *Front Pharmacol* **9**, 536 (2018).
7. Jiang, X.M., *et al.* Osimertinib (AZD9291) decreases programmed death ligand-1 in EGFR-mutated non-small cell lung cancer cells. *Acta Pharmacol Sin* **38**, 1512-1520 (2017).
8. Mittendorf, E.A., *et al.* PD-L1 expression in triple-negative breast cancer. *Cancer immunology research* **2**, 361-370 (2014).
9. Du, L., *et al.* β -Catenin induces transcriptional expression of PD-L1 to promote glioblastoma immune evasion. *J Exp Med* **217**(2020).
10. Simioni, C., *et al.* The AKT inhibitor MK-2206 is cytotoxic in hepatocarcinoma cells displaying hyperphosphorylated AKT-1 and synergizes with conventional chemotherapy. *Oncotarget* **4**, 1496-1506 (2013).

REVIEWER COMMENTS

Reviewer #1 (Remarks to the Author):

The authors have substantially revised their manuscript to the satisfaction of this reviewer.

Reviewer #2 (Remarks to the Author):

The authors have addressed most of the raised concerns during this round of revision.

Reviewer #3 (Remarks to the Author):

This is an interesting paper with plenty of worthwhile data. However, a problem remains with regards to GSK-3 and proof that this pathway is responsible for PDL-1 down-regulation.

1) Specifically, in their title that state the "WSX1...prevents CD8+ exhaustion"..there is no information on T-cell exhaustion and the markers used do not even include the basic TOX marker. The response that anti-TOX is not in the CYTOF panel and so is not examined is not adequate. FACs can be used for this. Also no other information is provided on T-cell exhaustion such as cytokine or proliferative effects..

2) Also the evidence that GSK-3 controls the process is based only on the use of lithium which is relatively non-specific and can target members of the ERK family and others. Basic transfection studies with inactive or constitutively active forms of GSK-3 would greatly strengthen the paper or the use of sh/siRNA or use GSK3-/- T-cells to determine whether GSK-3 truly accounts for the the effects of WSX1. Otherwise the blotting results on beta catenin are superficial...phosphorylation differences might occur but it does not mean that they are responsible for the effects of WSX1.

Point-by-Point Response to Reviewers' Comments

1) Specifically, in their title that state the "WSX1...prevents CD8+ exhaustion".there is no information on T-cell exhaustion and the markers used do not even include the basic TOX marker. The response that anti-TOX is not in the CYTOF panel and so is not examined is not adequate. FACs can be used for this. Also no other information is provided on T-cell exhaustion such as cytokine or proliferative effects.

Response 1:

Thanks for your valuable advice. In our previous submission, to clarify the effect of WSX1 on CD8+ T-cell exhaustion, we used T-cell exhaustion markers (PD-1, CTLA-4, and LAG-3) and T-cell functional markers (granzyme B, perforin, and Ki67). These markers have been widely used to indicate T-cell exhaustion status. However, TOX, as a novel biomarker, was not used in our first submission. As you suggested, in this revision, we included TOX as a T-cell exhaustion marker and analyzed the effects of WSX1 on IL-2 and IFN- γ production as well. Interestingly, TOX expression in CD8⁺ T-cells, which we analyzed by flow cytometry, was dramatically downregulated in the oncogene + WSX1 group, further supporting our notion that WSX1 relieved CD8+ T-cell exhaustion. Detailed information is shown in the revised Supplementary Figure S5.

2) Also the evidence that GSK-3 controls the process is based only on the use of lithium which is relatively non-specific and can target members of the ERK family and others. Basic transfection studies with inactive or constitutively active forms of GSK-3 would greatly strengthen the paper or the use of sh/siRNA or use GSK3^{-/-} T-cells to determine whether GSK-3 truly accounts for the the effects of WSX1. Otherwise the blotting results on beta catenin are superficial...phosphorylation differences might occur but it does not mean that they are responsible for the effects of WSX1.

Response 2:

In our previous submission, we used two approaches to verify that it is the GSK3 β pathway that is mostly responsible for WSX1-mediated PD-L1 reduction in HCC cells: administration of the GSK3 β inhibitor LiCl and blockade of GSK3 β -mediated PD-L1 phosphorylation by mutating GSK3 β phosphorylation sites on PD-L1. These two approaches independently validated our conclusion. Per the reviewer's suggestion, we included a third approach in this revision: GSK3 β shRNA transfection. As expected, GSK3 β knockdown significantly rescued WSX1-induced PD-L1 reduction. These data are shown in the revised Figure 6h-6j.

REVIEWERS' COMMENTS

Reviewer #3 (Remarks to the Author):

This remains an interesting paper with promise and of potential importance; however, it is still lacking the high standard expected to support their claims.

1) The new data on TOX1 shows significance but the level is reduced only by some 20%. This is a marginal effect compared to that expected for the 'reversal' of exhaustion as stated in the title and text. On page 12, the authors themselves state that exhaustion is shown by the "sustained expression of inhibitory receptors such as PD-1, LAG-3, Tim-3 and CTLA-428,20". The authors are missing the other key marker TIM3. In the tumor and chronic viral infection models, the standard for an exhausted T-cell is PD1hi, Tim3hi, Tox+. If the authors are going to make the claim then they need more rigorous data to prove the point. The effect on cytokines is encouraging. Reversal implies some 80-100% restoration of an effect. Their effects are therefore unlikely to reflect a true reversal of exhaustion. The claim also should be supported by a more nuanced definition of exhaustion as defined by the Wherry lab (see *Immunity* 2020 52(5):825-841). At this point, a more accurate interpretation would be a, 'partial reversion of CD8 T-cell dysfunction'

2) The results in Fig. 6h-j look promising but again key controls are missing. The authors need to show the actual down-regulation of GSK-3alpha and beta in an immuno-blot and whether both isoforms are affected. If they want to specifically focus on beta, they must explain their reasoning since LiCl affects both isoforms. They need to include a scrambled negative control..these are just basic controls. Most reports are required to outline the nature of the shRNAs. IT is also unclear why they are reliant on LiCl since reagents such as CHIR-99021 are much more specific inhibitors than this regent.

Point-by-point response to reviewer #3

Reviewer #3

General question from Reviewer #3: This remains an interesting paper with promise and of potential importance; however, it is still lacking the high standard expected to support their claims.

Response: Our primary discoveries (claims) are the following: (1) WSX1 is expressed in hepatocytes and downregulated in HCC. Elevated expression of WSX1 inhibits HCC incidence and progression *in vivo*, while knockout of WSX1 increases them; (2) WSX1 expression is positively associated with overall survival in HCC patients; (3) the IL-27-independent tumor-suppressive effect of WSX1 largely relies on CD8⁺ T-cell immune surveillance via reducing neoplastic PD-L1 expression and its associated CD8⁺ T-cell exhaustion; and (4) the underlying molecular mechanism is that WSX1 inhibits the PI3K δ /AKT/GSK3 β /PD-L1 pathway. It appears that Reviewer #3 agrees with the other reviewers on these main discoveries. The reviewer's main question regards a side claim (not our major claim) about the T-cell exhaustion biomarker(s) used. This will be addressed below.

Question (1) from Reviewer #3: The new data on TOX1 shows significance but the level is reduced only by some 20%. This is a marginal effect compared to that expected for the 'reversal' of exhaustion as stated in the title and text. On page 12, the authors themselves state that exhaustion is shown by the "sustained expression of inhibitory receptors such as PD-1, LAG-3, Tim-3 and CTLA-4,20". The authors are missing the other key marker TIM3. In the tumor and chronic viral infection models, the standard for an exhausted T-cell is PD1^{hi}, Tim3^{hi}, Tox⁺. If the authors are going to make the claim then they need more rigorous data to prove the point. The effect on cytokines is encouraging. Reversal implies some 80-100% restoration of an effect. Their effects are therefore unlikely to reflect a true reversal of exhaustion. The claim also should be supported by a more nuanced definition of exhaustion as defined by the Wherry lab (see *Immunity* 2020 52(5):825-841). At this point, a more accurate interpretation would be a 'partial reversion of CD8 T-cell dysfunction'.

Response (1): In our initial submission, we showed that WSX1 downregulates PD-L1 expression in HCC cells and reduces the proportion of exhausted CD8⁺ T cells. The exhausted CD8⁺ T cells were characterized by high and co-expression of widely used T-cell exhaustion markers (PD-1, CTLA-4, and LAG-3). None of the reviewers, including Reviewer 3, raised any issues regarding our results and conclusions at that point. After the first revision, Reviewer 3 requested that we study TOX1 because Reviewer 3 considered PD-1, CTLA-4, and LAG-3 to be inadequate markers of T-cell exhaustion, even though these biomarkers have been used by hundreds of labs in thousands of published papers. Though it was not entirely relevant to our major discoveries and our research goals, we added an examination of TOX1 expression in the second revision out of respect to this reviewer's comments.

For the evaluation of TOX1 expression, as shown in Supplementary Figure S5c and S5d, WSX1 decreased the proportion of TOX1⁺ CD8⁺ T cells from 51.2% \pm 8.002% to 29.433% \pm 3.011%, an increase not of 20%, as Reviewer 3 states above, but of 40%. The reason for the misunderstanding might be that we divided the Y axis into 2 segments to show both the CTLA-4⁺ and CTLA-4⁻ CD8⁺ T cells.

Reviewer 3 also criticized that we did not include TIM3, citing a paper that a standard set of T-cell exhaustion biomarkers include TIM3. In fact, no consistent standard exists for the pattern of inhibitory-receptor expression characterizing exhausted T cells in cancer. More and more researchers (including the authors of the cited paper *Immunity* 2020; 52(5):825-841 by reviewer 3) believe that exhausted T cells are a heterogeneous population with T-cell subsets in different developmental hierarchies and stages; during exhaustion, loss of T-cell function occurs in a hierarchical manner. The pattern of inhibitory-receptor coexpression is thus dynamic, diverse, and associated with the severity and character of dysfunction. Dysfunctional tumor-specific T cells have also been reported to co-express PD-1 and LAG-3 without TIM3. Therefore, we believe that the markers used in our study—the T-cell exhaustion markers (PD-1, CTLA-4, and LAG-3), T-cell functional markers (granzyme B, perforin, IL-2, IFN- γ , and Ki67), and TOX—while not perfect, are sufficient to characterize CD8⁺ T-cell exhaustion. Missing the marker TIM3 does not affect our conclusions. Of note, only a few TIM3⁺ CD8⁺ T cells were detected in our spontaneous HCC mouse models (as shown below), suggesting that TIM3 may not play an important role in the CD8⁺ T-cell exhaustion observed in our models. *Finally, we did not state that “WSX1 reversed CD8 T-cell dysfunction” in our manuscript, as stated by reviewer 3. Instead, our notion is that WSX1 reduced the number of exhausted CD8 T-cells (relieved CD8 T-cell exhaustion), as shown on page 6 of the main text.*

Question (2): The results in Fig. 6h-j look promising but again key controls are missing. The authors need to show the actual down-regulation of GSK-3 α and beta in an immuno-blot and whether both isoforms are affected. If they want to specifically focus on beta, they must explain their reasoning since LiCl affects both isoforms. They need to include a scrambled negative control. these are just basic controls. Most reports are required to outline the nature of the shRNAs. IT is also unclear why they are reliant on LiCl since reagents such as CHIR-99021 are much more specific inhibitors than this reagent.

Response (2): The immunoblotting results are shown below. No significant effect of GSK3 β shRNAs on GSK3 α protein levels was observed.

In our study, we included scrambled shRNAs as a negative control for GSK3 β shRNAs and a solvent control as a negative control for LiCl. The reason we didn't show the group with scrambled shRNAs separately is that no significant differences in the proportions of PD-L1⁺ HCC cells among blank control, solvent control, and scrambled shRNAs were found. The "control" in Figure 6h and 6i represents a representative level for solvent control and scrambled shRNAs.

LiCl has been widely used for the study of GSK3 β activity. As stated by the reviewer, considering the potential effect of LiCl on GSK3 α activity, we cannot rule out the possibility that GSK3 α might also play a part in regulating PD-L1 expression. However, so far, no study has reported the regulation of PD-L1 by GSK3 α , while a great deal of evidence has supported the critical role of GSK3 β . Therefore, in this study we focused on GSK3 β . The full story underlying the regulation of WSX1 and PD-L1 might need another several manuscripts to uncover.